# Efficiently Learning One Hidden Layer Neural Networks From Queries

**Sitan Chen***
sitanc@mit.edu
MIT

**Adam R. Klivans**[†]
klivans@cs.utexas.edu
UT Austin

**Raghu Meka**[‡]
raghum@cs.ucla.edu
UCLA

## Abstract

Model extraction attacks have renewed interest in the classic problem of learning neural networks from queries. This work gives the first polynomial-time algorithm for learning one hidden layer neural networks provided black-box access to the network. Formally, we show that if $F$ is an arbitrary one hidden layer neural network with ReLU activations, there is an algorithm with query complexity and running time that is polynomial in all parameters that outputs a network $F'$ achieving low square loss relative to $F$ with respect to the Gaussian measure. While a number of works in the security literature have proposed and empirically demonstrated the effectiveness of certain algorithms for this problem, ours is the first with fully polynomial-time guarantees of efficiency for worst-case networks (in particular our algorithm succeeds in the overparameterized setting).

## 1 Introduction

The problem of learning neural networks given random labeled examples continues to be a fundamental algorithmic challenge in machine learning theory. There has been a resurgence of interest in developing algorithms for learning one hidden layer neural networks, namely functions of the form $F(x) = \sum_{i=1}^{k} s_i \sigma(\langle w_i, x \rangle - b_i)$ where $w_i \in \mathbb{R}^d$, $s_i \in \{\pm 1\}$, $b_i \in \mathbb{R}$ are the unknown parameters. Here we consider the ubiquitous setting where $\sigma$ is the ReLU activation, the input distribution on $x$ is Gaussian, and the goal is to output a ReLU network $F'$ with $\mathbb{E}[(F(x) - F'(x))^2] \leq \epsilon$.

Despite much recent work on this problem [JSA15, ZSJ+17, GKM18, GLM18, GKLW18, ZYWG19, BJW19, DKKZ20, CKM20, LMZ20], obtaining a polynomial-time algorithm remains open. All known works take additional assumptions on the unknown weights and coefficients or do not run in polynomial-time in all the important parameters. In fact, there are a number of lower bounds, both for restricted models of computation like correlational statistical queries [GGJ+20, DKKZ20] as well as under various average-case assumptions [DV20, DV21], that suggest that a truly polynomial-time algorithm may be impossible to achieve.

In this work, we strengthen the learner by allowing it query access to the unknown network. That is, the learner may select an input $x$ of its choosing and receive the output value of the unknown network $F(x)$. Our main result is the first polynomial-time algorithm for learning one hidden layer ReLU networks with respect to Gaussian inputs when the learner is given query access:

**Theorem 1.1.** *Let $F(x) = \sum_{i=1}^{k} s_i \sigma(\langle w_i, x \rangle - b_i)$ with $\|w_i\|_2 \leq R$, $s_i \in \{\pm 1\}$, and $b_i \leq B$. Given black-box access to $F$, there exists a (randomized) algorithm that will output with probability at least*

---

*This work was supported in part by an NSF CAREER Award CCF-1453261 and NSF Large CCF-1565235.

[†]Supported by NSF awards AF-1909204, AF-1717896, and the NSF AI Institute for Foundations of Machine Learning (IFML).

[‡]Supported by NSF CAREER Award CCF-1553605.

35th Conference on Neural Information Processing Systems (NeurIPS 2021).

$1 - \delta$, *a one hidden layer network $F'$ such that $\mathbb{E}_{x \sim \mathcal{N}(0, \mathrm{Id})}[(F(x) - F'(x))^2] \leq \epsilon$. The algorithm has query complexity and running time that is polynomial in $d, B, R, k, 1/\epsilon$, and $\log(1/\delta)$.*

In light of the aforementioned lower bounds [GGJ$^+$20, DKKZ20, DV20, DV21], Theorem 1.1 suggests there is a separation between learning from random samples and learning from queries in the context of neural networks.

In addition to being a naturally motivated challenge in computational learning theory, the question of learning neural networks from queries has also received strong interest from the security and privacy communities in light of so-called *model extraction attacks* [TJ$^+$16, MSDH19, PMG$^+$17, JCB$^+$20, RK20, JWZ20]. These attacks attempt to reverse-engineer neural networks found in publicly deployed real-world systems. Since the target network is publicly available, an attacker therefore obtains black-box query access to the network — exactly the learning model we are working in. Of particular interest is the work of Carlini et al. [CJM20] (see also [JCB$^+$20]), who gave several heuristics for learning deep neural networks given black-box access (they also empirically verified their results on networks trained on MNIST). Another relevant work is that of [MSDH19] which gave theoretical guarantees for the problem we study under quite strong separation/linear independence assumptions on the weight vectors; in fact they are able to *exactly* recover the network, but in the absence of such strong assumptions this is impossible (see Section 1.2). Since these papers bear strong parallels with our techniques, in Section 1.2 we provide a more detailed description of why these approaches break down in our setting and highlight the subtleties that we address to achieve the guarantee of Theorem 1.1.

## 1.1 Our Approach

Here we describe our approach at a high level. Fix an unknown one hidden-layer network $F(x) = \sum_{i=1}^{k} s_i \sigma(\langle w_i, x \rangle - b_i)$ with weight vectors $w_i \in \mathbb{R}^d$, signs $s_i \in \{\pm 1\}$, and bias terms $b_i \in \mathbb{R}$. Consider a line $L = \{x_0 + t \cdot v\}_{t \in \mathbb{R}}$ for $x_0, v \in \mathbb{R}^d$ and the *restriction* $F|_L(t) \triangleq F(x_0 + t \cdot v)$ given by

$$F|_L(t) = \sum_{i=1}^{k} s_i \sigma\left(\langle w_i, x_0 \rangle - b_i + t \langle w_i, v \rangle\right).$$

Note that as a univariate function, $F|_L(t)$ is a piecewise-linear function. We call a point $t \in \mathbb{R}$ a *critical point* of $F|_L$ if the slope of the piecewise-linear function changes at $t$. Our starting point is that if we can identify the critical points of $F|_L$, then we can use estimates of the gradient of $F$ at those critical points to estimate the weights of the hidden units in $F$. More concretely, suppose we have identified a critical point $t_i$ of $F|_L$ for a line $L$. Now, back in the $d$-dimensional space, if we look at a sufficiently small neighborhood of the point $x' = x_0 + t_i \cdot v$, then the set of activated units around $x'$ changes by exactly one (in fact, by this reasoning, for a random $L$, $F|_L$ will have one unique critical point for each neuron in $F$, provided no two neurons are exact multiples of each other). We can exploit this to retrieve the $w_i$'s and $b_i$'s by finite differencing: for each such $x'$, compute $(F(x' + \delta) - F(x'))/\|\delta\|$, for several sufficiently small perturbations $\delta \in \mathbb{R}^d$, and this will recover $s_i w_i$ (see Algorithm 2 and 4.9 below for the details). We can recover $s_i b_i$ in a similar fashion (see Algorithm 1). We remark that the approach we have described appears to be quite similar to that of [CJM20, JCB$^+$20, MSDH19].

However, the above description leaves out the following key interrelated challenges for getting provable guarantees for arbitrary networks:

1. How do we identify the critical points of $F|_L$ for a given line $L$?

2. If we make no assumptions on how well-separated the weight vectors are, it is actually impossible to recover all of the weight vectors and biases in a bounded number of queries. Using the neurons that we do approximately recover, is it possible to piece them together to approximate $F$?

A rigorous solution to these issues turns out to be quite tricky and technically involved. For instance, what if some critical points are arbitrarily close to each other (imagine the piecewise linear function being a tiny peak of exponentially small in $d$ width on the real line) on the line $F|_L$. In this case, identifying them would not be possible efficiently. We develop several structural results about cancellations in ReLU networks (culminating in Lemma 3.9) that show that such a situation only

arises in degenerate cases where some sub-network of $F$ contains a cluster of many similar neurons (we formalize the notion of similarity in Definition 3.1). We show in Section 3 that much smaller networks can approximate these sub-networks. On the other hand, for pairs of neurons which are sufficiently far from each other, the critical points on a random line $L$ will be well-separated.

With the structural results in hand, roughly speaking it suffices to identify one "representative" neuron for every cluster of similar neurons in $F$. To do this, we discretize $L$ into intervals of equal length and look at differences between gradients/biases of $F$ at the midpoints of these intervals. The key result leveraging our structural results is Theorem 4.8 which shows that the set of all such differences comprises a rich enough collection of neurons that we can assemble some linear combination of them that approximates $F$. To find a suitable linear combination, we run linear regression on the features computed by these neurons (see Section 4.4).

## 1.2 Comparison to Previous Approaches

Our general approach of looking for critical lines along random restrictions of $F|_L$ is also the approach taken in the empirical works of [CJM20, JCB$^+$20] and in the theoretical work of [MSDH19], but we emphasize that there are a variety of subtleties that arise because we are interested in learning *worst-case* networks from queries. In contrast, the empirical works consider trained networks arising in practice, while [MSDH19] makes a strong assumption that the weight vectors $w_i$ are *linearly independent* and that they are angularly separated. Note that such assumptions cannot hold in the practically relevant setting where $F$ is overparameterized.

To give a sense for the issues that these existing techniques run up against when it comes to worst-case networks, imagine that the one-dimensional piecewise linear function given by the restriction $F|_L$ were simply a "bump," that is, it is zero over most of the line except in a small interval $[a, a+\delta]$. For instance, this would arise in the following example:

**Example 1.2.** *Consider the one-dimensional one hidden layer network $F : \mathbb{R} \to \mathbb{R}$ given by $F(x) = \sigma(x-a) + \sigma(x-a-\delta) - \sigma(2x-2a-\delta)$. This function looks like the zero function except over the interval $[a, a+\delta]$, where it looks like a small "bump."*

The proposal in the works mentioned above is to run a binary search to find a critical point. Namely, they initialize to some large enough interval $[-\tau, \tau]$ in $L$ and check whether the gradient at the left or right endpoint differs from the gradient at the midpoint $t_{\mathsf{mid}} = 0$. If the former, then they restrict the search to $[-\tau, t_{\mathsf{mid}}]$ and recurse until they end up with a sufficiently small interval, at which point they return the midpoint of that interval as an approximation to a critical point. The intuition is that $n$ steps of binary search suffice to identify a critical point up to $n$ bits of precision.[4]

It is clear, however, from the "bump" example that such a binary search procedure is doomed to fail: if the bump does not occur in the middle of the starting interval $[-\tau, \tau]$, then at the outset, we don't know which half to recurse on because the gradients at the endpoints and the gradient at the midpoint are all zero! Indeed, to locate the bump, it is clear that we need to query a number of points which is at least inverse in the width of the bump to even find an input to the network with nonzero output.

We remark that this is in stark contrast to related active learning settings where the ability to query the function can sometimes yield *exponential savings* on the dependence on $1/\epsilon$ (see e.g. [Han09]). Indeed, if we took the width of the bump to be of size $\mathrm{poly}(\epsilon)$ to make it $\epsilon$-far in square distance from the zero function, we would still need $\mathrm{poly}(1/\epsilon)$, rather than $\log(1/\epsilon)$, queries to locate the bump and distinguish the funcction from the zero function.

The core reason binary search fails for arbitrary networks is that there can be many disjoint linear pieces of $F|_L$ which all have the same slope. Indeed, because the gradient of $F$ at a given point is some linear combination of the weight vectors, if $F$ is overparameterized so that there are many linear dependencies among the weight vectors, it is not hard to design examples like Example 1.2 where there may be many repeated slopes on any given random restriction of $F$ to a line.

Apart from these technical issues that arise in the worst-case setting we consider and not in previous empirical or theoretical works on model extraction, we also emphasize that our results are the first

---

[4]We remark that the refinement of binary search given in [CJM20, JCB$^+$20] can speed this up to $O(1)$ steps for networks that arise in practice, but the issue that we describe poses a challenge for their technique as well.

theoretical guarantees for learning general one hidden-layer ReLU networks with bias terms in any learning model, including PAC learning from samples. As will quickly become evident in the sequel, biases introduce many technical hurdles in their own right. To our knowledge, the only other theoretical guarantees for learning networks with biases are the work of [JSA15] which considers certain activations with nonzero second/third-order derivatives, precluding the ReLU, and the recent work of [ATV21] which considers ReLU networks whose weight vectors are well-conditioned.

**Remark 1.3.** *As discussed above, the "bump" construction in Example 1.2 shows that unlike in related active learning contexts,* $\text{poly}(1/\epsilon)$ *dependence is necessary in our setting. In fact this example also tells us another somewhat counterintuitive fact about our setting. Naively, one might hope to upgrade Theorem 1.1 by replacing the dependence on the scaling parameters $R, B$ with one solely on the $L_2$ norm of the function. To see why this is impossible, consider scaling the function in Example 1.2 by a factor of $\delta^{-3/2}$ to have unit $L_2$ norm. To learn $F$, we have to figure out the location of the bump, but this requires $\Omega(1/\delta)$ queries, and $\delta$ can be taken to be arbitrarily small.*

**Limitations and Societal Impact**   The line of work on learning neural networks from black-box access does pose a risk, for instance, that proprietary models offered via publicly-hosted APIs may be stolen by attackers who are only able to interact with the models through queries. The attackers might then use the extracted parameters of the model to learn sensitive information about the data the model was trained on, or perhaps to construct adversarial examples. That being said, understanding why this query learning problem can be easy for theorists is the first step towards building provable safeguards to ensure that it is hard for actual attackers. For instance, it is conceivable that one can prove information-theoretic or computational hardness for such problems if appropriate levels and kinds of noise are injected into the responses to the queries. Furthermore, query complexity lower bounds can inform how many accesses an API should allow any given user.

## 2   Preliminaries

**Notation**   We let $\mathbb{S}^{d-1}$ denote the unit sphere in $\mathbb{R}^d$. Let $e_j$ denote the $j$-th standard basis vector in $\mathbb{R}^d$. Given vectors $u, v$, let $\angle(u, v) \triangleq \arccos\left(\frac{\langle u,v \rangle}{\|u\|\|v\|}\right)$. Given a matrix $M$, let $\|M\|_{\mathsf{op}}$ and $\|M\|_F$ denote the operator and Frobenius norms respectively. Given a function $h$ which is square-integrable with respect to the Gaussian measure, we will use $\|h\|$ to denote $\mathbb{E}_{x\sim\mathcal{N}(0,\mathrm{Id})}[h(x)^2]^{1/2}$. Given a collection of indices $S \subseteq \mathbb{Z}$, we say that $i, j \in S$ are *neighboring* if there does not exist $i < \ell < j$ for which $\ell \in S$.

**Definition 2.1.** *A* neuron *is a pair $(v, b)$ where $v \in \mathbb{R}^d$ and $b \in \mathbb{R}$; it corresponds to the function $x \mapsto \sigma(\langle v, x \rangle - b)$, which we sometimes denote by $\sigma(\langle v, \cdot \rangle - b)$.*

**Definition 2.2.** *Given a line $L \subset \mathbb{R}^d$ parameterized by $L = \{x_0 + t \cdot v\}_{t\in\mathbb{R}}$, and a function $F : \mathbb{R}^d \to \mathbb{R}$, define the* restriction *of $F$ to $L$ by $F|_L(t) \triangleq F(x_0 + t \cdot v)$.*

**Definition 2.3.** *Given a line $L \subset \mathbb{R}^d$ and a restriction $F|_L$ of a piecewise linear function $F : \mathbb{R}^d \to \mathbb{R}$ to that line, the* critical points *of $F$ are the points $t \in \mathbb{R}$ at which the slope of $F|_L$ changes.*

We will need the following standard facts about the concentration of the norm and individual entries of a Gaussian vector.

**Fact 2.4.** *Given Gaussian vector $h \sim \mathcal{N}(0, \boldsymbol{\Sigma})$, $\Pr\left[\|h\| \geq O(\|\boldsymbol{\Sigma}^{1/2}\|_{\mathsf{op}}(\sqrt{r} + \sqrt{\log(1/\delta)}))\right] \leq \delta$, where $r$ is the rank of $\boldsymbol{\Sigma}$.*

**Fact 2.5.** *For covariance matrix $\boldsymbol{\Sigma} \in \mathbb{R}^{m\times m}$, given $h \sim \mathcal{N}(0, \boldsymbol{\Sigma})$ we have that $|h_i| \leq O\left(\sqrt{\boldsymbol{\Sigma}_{i,i}}\sqrt{\log(m/\delta)}\right)$ for all $i \in [m]$ with probability at least $1 - \delta$.*

We will also need anti-concentration of the norm of a Gaussian vector and the individual entries of a random unit vector.

**Lemma 2.6.** *There is an absolute constant $C > 0$ such that given any Gaussian vector $h \sim \mathcal{N}(\mu, \boldsymbol{\Sigma})$, $\Pr\left[\|h\| \geq \sqrt{\nu}\|\boldsymbol{\Sigma}\|_F^{1/2}\right] \geq 1 - C\sqrt{\nu}$.*

**Lemma 2.7.** *For random $v \in \mathbb{S}^{d-1}$, $\Pr\left[|v_1| < \frac{\delta}{2\sqrt{d}+O(\sqrt{\log(1/\delta)})}\right] \leq \delta$.*

# 3 ReLU Networks with Cancellations

In the following section we prove several general results about approximating one hidden-layer networks with many "similar" neurons by much smaller networks, deferring proofs to the supplementary material.

## 3.1 $(\Delta, \alpha)$-Closeness of Neurons

We first formalize a notion of geodesic closeness between two neurons and record some useful properties. This notion is motivated by Lemma 4.4 where we study the critical points of random restrictions of one hidden-layer networks.

**Definition 3.1.** *Given $v, v' \in \mathbb{R}^d$ and $b, b' \in \mathbb{R}$, we say that $(v, b)$ and $(v', b')$ are $(\Delta, \alpha)$-close if the following two conditions are satisfied:*

1. *$|\sin \angle(v, v')| \leq \Delta$*

2. *$\|bv' - b'v\| \leq \alpha \|v\| \|v'\|$.*

*Note that this is a measure of angular closeness between $(v, b), (v', b') \in \mathbb{R}^{d+1}$. For instance, if $(v, b) = (\lambda v^*, \lambda b^*)$ and $(v', b') = (\lambda' v^*, \lambda' b^*)$ for some $(v^*, b^*)$, then $(v, b)$ and $(v', b')$ are $(0, 0)$-close.*

We first collect some elementary consequences of closeness. The following intuitively says that if we scale two $(\Delta, \alpha)$-neurons to have similar norm, then their biases will be close.

**Lemma 3.2.** *If $(v, b)$ and $(v', b')$ are $(\Delta, \alpha)$-close, and $v = \gamma v' + v^\perp$ for $v^\perp$ orthogonal to $v'$, then $|\gamma b' - b| \leq \alpha \|v\|$.*

Note that when two neurons are $(\Delta, \alpha)$-close, their weight vectors are either extremely correlated or extremely anti-correlated. In fact, given a collection of neurons that are all pairwise close, they will exhibit the following "polarization" effect.

**Lemma 3.3.** *Suppose $\Delta < \sqrt{2}/2$. If $(v_1, b_1), \ldots (v_k, b_k)$ are all pairwise $(\Delta, \alpha)$-close for some $\alpha > 0$, then there is a partition $[k] = S_1 \sqcup S_2$ for which $\langle v_i, v_j \rangle \geq 0$ for any $i \in S_1, j \in S_1$ and $i \in S_2, j \in S_2$, and for which $\langle v_i, v_j \rangle < 0$ for any $i \in S_1, j \in S_2$ and $i \in S_2, j \in S_1$.*

In the rest of the paper we will take $\Delta$ to be small, so Lemma 3.3 will always apply. As such, it will be useful to define the following terminology:

**Definition 3.4.** *Given $(v_1, b_1), \ldots, (v_k, b_k)$ which are all pairwise-close, we will call the partition $S_1 \sqcup S_2$ given in Lemma 3.3 the* orientation induced by $\{(v_i, b_i)\}$.

We note that $(\Delta, \alpha)$-closeness satisfies the triangle inequality.

**Lemma 3.5.** *If $(v_1, b_1)$ and $(v_2, b_2)$ are $(\Delta, \alpha)$-close, and $(v_2, b_2)$ and $(v_3, b_3)$ are $(\Delta', \alpha')$-close, then $(v_1, b_1)$ and $(v_3, b_3)$ are $(\Delta + \Delta', 2\alpha + 2\alpha')$-close.*

## 3.2 Merging Neurons

We prove that a one hidden-layer network where all neurons are $(\Delta, \alpha)$-close to some neuron can be approximated by at most two neurons:

**Lemma 3.6.** *[See supplementary material.] Given $F(x) = \sum_{i=1}^k s_i \sigma(\langle w_i, x \rangle - b_i)$ for $s_i \in \{\pm 1\}$ and $(v^*, b^*) \in \mathbb{R}^d \times \mathbb{R}$ for which $(w_i, b_i)$ is $(\Delta, \alpha)$-close to $(v^*, b^*)$ for all $i \in [k]$, there exist coefficients $a^+, a^- \in \mathbb{R}$ for which*

$$\mathop{\mathbb{E}}_{x \sim \mathcal{N}(0, Id)} \left[ \left( F(x) - a^+ \sigma(\langle v^*, x \rangle - b^*) - a^- \sigma(\langle -v^*, x \rangle + b^*) \right)^2 \right] \leq O(k^2 (\Delta^{2/5} + \alpha^2)) \|v^*\|^2.$$

*Furthermore, we have that*

$$|a^+| \|v^*\|, |a^-| \|v^*\| \leq \sum_i \|w_i\| \qquad and \qquad |a^+ b^*|, |a^- b^*| \leq \alpha \sum_i \|w_i\| + \sum_i |b_i|.$$

Note that Lemma 3.6 requires the existence of a neuron $(v^*, b^*)$ which is close to all neurons $\{(v_i, b_i)\}$. In our algorithm, we will not have access to $(v^*, b^*)$ but rather to some linear combination of the neurons $\{(v_i, b_i)\}$. We show that provided this linear combination is not too small in norm, it will also be close to all the neurons $\{(v_i, b_i)\}$.

**Lemma 3.7.** *Suppose we have vectors* $v_1, \ldots, v_m, v^* \in \mathbb{R}^d$, *biases* $b_1, \ldots, b_m, b^* \in \mathbb{R}$ *for which every* $(v_i, b_i)$ *is* $(\Delta, \alpha)$-*close to* $(v^*, b^*)$. *Then for any* $s_1, \ldots, s_m \in \{\pm 1\}$, *if we define* $v \triangleq \sum_{i=1}^m s_i v_i$ *and* $b \triangleq \sum_{i=1}^m s_i b_i$, *then* $(v, b)$ *is* $(\Delta m, \alpha \sum_i \|v_i\|/\|v\|)$-*close to* $(v^*, b^*)$.

This presents an issue however: what if the linear combination of neurons that we get access to in our eventual algorithm has small norm, in which case Lemma 3.7 is not helpful? It turns out this linear combination takes a very specific form (see the vector in (1)), and we argue that if it is indeed small, then the underlying network we are trying to approximate will be close to a linear function! Formally we show:

**Lemma 3.8.** *Suppose* $(v_1, b_1), \ldots, (v_m, b_m)$ *are pairwise* $(\Delta, \alpha)$-*close, and let* $[m] = S_1 \sqcup S_2$ *denote the orientation induced by them (see Definition 3.4). Define* $B \triangleq \max_i \|b_i\|$ *and* $R \triangleq \max_i \|v_i\|$. *If signs* $s_1, \ldots, s_m \in \{\pm 1\}$ *satisfy*

$$\|\sum_{i \in S_1} s_i v_i - \sum_{i \in S_2} s_i v_i\| \leq (\Delta R)^{2/9}, \tag{1}$$

*then for the network* $F(x) \triangleq \sum_i s_i \sigma(\langle v_i, x \rangle - b_i)$, *there exists an affine linear function* $\ell(x) : \mathbb{R}^d \to \mathbb{R}$ *for which*

$$\mathop{\mathbb{E}}_{x \sim \mathcal{N}(0, Id)} \left[ (F(x) - \ell(x))^2 \right] \leq \text{poly}(k, R, B) \cdot (\alpha^{1/2} + \Delta^{2/9})$$

*where* $\ell \triangleq \langle w^*, \cdot \rangle - b^*$ *satisfies*

$$\|w^*\| \leq \sum_i \|v_i\| \qquad \text{and} \qquad |b^*| \leq \sum_i \|b_i\|.$$

### 3.3 Putting Everything Together

Putting Lemmas 3.6, 3.7, and 3.8 together, we conclude that networks whose hidden units are pairwise $(\Delta, \alpha)$-close can either be approximated by a particular size-two network, or by *some* affine linear function:

**Lemma 3.9.** *Suppose* $(v_1, b_1), \ldots, (v_k, b_k)$ *are pairwise* $(\Delta, \alpha)$-*close, and let* $[k] = S_1 \sqcup S_2$ *denote the orientation induced by them (see Definition 3.4). Define* $B \triangleq \max_i \|b_i\|$ *and* $R \triangleq \max_i \|v_i\|$. *Let* $s_1, \ldots, s_m \in \{\pm 1\}$.

*Define* $F(x) = \sum_i s_i \sigma(\langle v_i, x \rangle - b_i)$, $v^* = \sum_{i \in S_1} s_i v_i - \sum_{i \in S_2} s_i v_i$, *and* $b^* = \sum_{i \in S_1} s_i b_i - \sum_{i \in S_2} s_i b_i$. *At least one of the following holds:*

1. *There is an affine linear function* $\ell : \mathbb{R}^d \to \mathbb{R}$ *for which* $\|F - \ell\|^2 \leq \text{poly}(k, R, B) \cdot (\alpha^{1/2} + \Delta^{2/9})$.

2. *There exist coefficients* $a^+, a^- \in \mathbb{R}$ *for which* $G(x) \triangleq a^+ \sigma(\langle v^*, x \rangle - b^*) - a^- \sigma(\langle -v^*, x \rangle + b^*)$ *satisfies* $\|F - G\|^2 \leq \text{poly}(k, R, B) \cdot (\Delta^{2/5} + \alpha^2 \Delta^{-4/9})$.

*Proof.* By assumption, every $(v_i, b_i)$ is $(\Delta, \alpha)$-close to $(v_1, b_1)$. By Lemma 3.7 we get that for $(v^*, b^*)$ defined in the lemma statement, $(v_1, b_1)$ is $(\Delta k, \alpha m R / \|v^*\|)$-close to $(v^*, b^*)$.

If $\|v^*\| \geq (\Delta R)^{2/9}$, then we conclude that $(v_1, b_1)$ is $(\Delta k, \alpha m \Delta^{-2/9} R^{7/9})$-close to $(v^*, b^*)$, and by Lemma 3.6 we find that there is a choice of $a^+, a^-$ for which the function $G$ defined in the lemma statement satisfies $\|F - G\|^2 \leq O(k^4 R^2 (\Delta^{2/5} k^{2/5} + \alpha^2 m^2 \Delta^{-4/9} R^{14/9}))$ (note that we used $\|v^*\| \leq \sum_i \|v_i\| \leq kR$).

If $\|v^*\| \leq (\Delta R)^{2/9}$, then by Lemma 3.8 we find that there is an affine linear $\ell$ for which $\|F - \ell\|^2 \leq \text{poly}(k, R, B) \cdot (\alpha^{1/2} + \Delta^{2/9})$. □

# 4 Learning One Hidden Layer

In this section we give our algorithm for learning neural networks from queries. Throughout, we will suppose we have black-box query access to some unknown one-hidden layer neural network

$$F(x) \triangleq \sum_{i=1}^{k} s_i \sigma(\langle w_i, x \rangle - b_i), \tag{2}$$

where $s_i \in \{\pm 1\}$, $w_i \in \mathbb{R}^d$, $b_i \in \mathbb{R}$. Define the quantities $R \triangleq \max_i \|w_i\|$ and $B \triangleq \max_i \|b_i\|$; our bounds will be polynomial in these quantities, among others.

## 4.1 Critical Points of One-Hidden Layer Networks

In this section, we compute the critical points of restrictions of $F$ and argue that they are far apart along *random restrictions* unless if the corresponding neurons were close to begin with (in the sense of Definition 3.1).

First, we formalize the notion of a random restriction:

**Definition 4.1.** *A* Gaussian line $L$ *is a random line in $\mathbb{R}^d$ formed as follows: sample $x_0 \sim \mathcal{N}(0, Id)$ and Haar-random $v \in \mathbb{S}^{d-1}$ and form the line $L \triangleq \{x_0 + t \cdot v\}_{t \in \mathbb{R}}$.*

Here we compute the critical points along a restriction of $F$.

**Proposition 4.2.** *Given a line $L = \{x_0 + t \cdot v\}_{t \in \mathbb{R}}$, the restriction $F|_L(t) \triangleq F(x_0 + t \cdot v)$ is given by*

$$F|_L(t) = \sum_{i=1}^{k} s_i \sigma \left( \langle w_i, x_0 \rangle - b_i + t \langle w_i, v \rangle \right).$$

*This function has $k$ critical points, namely $t = -\frac{\langle w_i, x_0 \rangle - b_i}{\langle w_i, v \rangle}$ for every $i \in [k]$.*

*Proof.* The critical points of $F|_L$ are precisely the points $t$ at which a neuron changes sign. So the crticial point associated to the $i$-th neuron is the $t$ for which $\langle w_i, x_0 \rangle - b_i + t \langle w_i, v \rangle = 0$, from which the claim follows. $\square$

We can show that these critical points are not too large, unless the norm of the corresponding weight vector is small. The reason for the latter caveat is that, e.g., if one took the one-dimensional neuron $\sigma(\epsilon z - b)$ for $b$ fixed and $\epsilon \to 0$, the $z$ at which it changes sign tends to $\infty$).

**Lemma 4.3.** *With probability at least $1 - \delta$ over the randomness of Gaussian line $L$, we have that $|t_i| \lesssim \frac{k(\sqrt{d} + \sqrt{\log(1/\delta)})}{\delta \|w_i\|} + k \left( \sqrt{d} + \sqrt{\log(1/\delta)} \right) \sqrt{\log(k/\delta)}$ for every critical point $t_i$ of $F|_L$.*

Fix a separation parameter $\Delta > 0$ which we will tune in the sequel. We show that along Gaussian lines $L$, $F|_L$'s critical points are well-separated except for those corresponding to neurons which are $(\Delta, \alpha)$-close.

**Lemma 4.4.** *There is an absolute constant $c > 0$ for which the following holds. Given Gaussian line $L$, with probability at least $1 - \delta$ we have: for any pair of $i, j$ for which $(w_i, b_i)$ and $(w_j, b_j)$ are not $(\Delta, c\Delta \sqrt{\log(k/\delta)})$-close, the corresponding critical points are at least $\Omega \left( \frac{\Delta \delta^2}{k^4 \left( \sqrt{d} + \sqrt{\log(k/\delta)} \right)} \right)$- apart.*

## 4.2 Line Search and Existence Theorem

At a high level, our algorithm works by searching along $F|_L$, partitioning $L$ into small intervals, and computing differences between the gradients/biases of $F$ at the midpoints of these intervals. The primary structural result we must show is that there exists enough information in this set of differences to reconstruct $F$ up to small error.

As we will be working with partitions of lines, it will be convenient to define the following notation:

**Definition 4.5.** *Given line $L \subset \mathbb{R}^d$ and finite interval $I \subseteq \mathbb{R}$ corresponding to a segment $\mathcal{I} \subset L$, let $\nabla_L(I)$ denote the gradient of $F$ at the midpoint of $\mathcal{I}$. For $t_{\mathsf{mid}} \in \mathbb{R}$ the midpoint of $I$, define $b_L(I) \triangleq F|_L(t_{\mathsf{mid}}) - (F|_L)'(t_{\mathsf{mid}}) \cdot t_{\mathsf{mid}}$. Intuitively, this is the "y-intercept" of the linear piece of $F|_L$ that contains $t_{\mathsf{mid}}$. When $L$ is clear from context, we will drop subscripts and denote these objects by $\nabla(I)$ and $b(I)$.*

**Definition 4.6.** *Given line $L \subset \mathbb{R}^d$ and length $s > 0$, let $\{t_i\}$ denote the critical points of $F|_L$, and let $G_L^+(s) \subseteq [k]$ (resp. $G_L^-(s)$) denote the set of indices $a \le i \le b$ for which $t_{i+1} - t_i \ge s$ (resp. $t_i - t_{i-1} \ge s$). Let $G_L^*(r) \triangleq G_L^+(r) \cap G_L^-(r)$.*

The following observation motivates Definition 4.6:

**Observation 4.7.** *Given line $L \subset \mathbb{R}^d$, let $\{t_i\}$ denote the critical points of $F|_L$. Let $I_1, \dots, I_m$ be a partition of some interval $I$ into pieces of length $r$, and for $t_i \in I$ let $\ell(i)$ denote the index of the interval containing $I$. Then for any $i \in G_L^+(2r)$ (resp. $i \in G_L^-(2r)$), $I_{\ell(i)+1}$ is entirely contained within $[t_i, t_{i+1}]$ (resp. $I_{\ell(i)-1}$ is entirely contained within $[t_{i-1}, t_i]$). In particular, $I_{\ell(i)-1}$ and $I_{\ell(i)+1}$ are linear pieces of $F|_L$.*

The following is the main result of this section. At a high level, it says if we partition a random line in $\mathbb{R}^d$ into sufficiently small intervals and can compute the gradient of $F$ at the midpoint of each interval, then we can produce a collection of neurons which can be used to approximate $F$.

**Theorem 4.8.** *For any $\epsilon, \delta > 0$, define*

$$r \triangleq O\left(\frac{\Delta\delta^2}{k^4\left(\sqrt{d} + \sqrt{\log(k/\delta)}\right)}\right) \tag{3}$$

$$\tau \triangleq kr + \Theta\left(\frac{k(\sqrt{d} + \sqrt{\log(1/\delta)})}{\delta\|w_i\|} + k\left(\sqrt{d} + \sqrt{\log(1/\delta)}\right)\sqrt{\log(k/\delta)}\right). \tag{4}$$

*Partition the interval $[-\tau, \tau]$ into intervals $I_1, \dots, I_m$ of length $r$.*

*Let $L$ be a Gaussian line, and let $\mathcal{S}$ denote the set of all $m(m-1)$ pairs $(w, b)$ obtained by taking distinct $i, j \in [k]$ and forming $(\nabla_L(I_i) - \nabla_L(I_j), b_L(I_i) - b_L(I_j))$. There exist $\{\pm 1\}$-valued coefficients $\{a_{w,b}\}_{(w,b) \in \mathcal{S}}$, vector $w^*$, and $b^* \in \mathbb{R}$ for which*

$$\left\|F - \sum_{(w,b) \in \mathcal{S}} a_{w,b} \cdot \sigma(\langle w, \cdot \rangle - b) - \langle w^*, \cdot \rangle - b^*\right\| \le \epsilon + \mathfrak{P}_{k,R,B,\log(1/\delta)} \cdot \Delta^{2/9}.$$

*for $\mathfrak{P}_{k,R,B,\log(1/\delta)}$ some absolute constant that is polynomially large in $k, R, B, \log(1/\delta)$. Furthermore, we have that*

$$\|a_{w,b} \cdot w\| \le kR \qquad and \qquad |a_{w,b} \cdot b| \le c\Delta k^2 R\sqrt{\log(k/\delta)} + kB \tag{5}$$

$$\|w^*\| \le kR \qquad and \qquad |b^*| \le kB$$

### 4.3 Gradient and Bias Oracles

It remains to implement oracles to compute $b_L(I)$ and $\nabla_L(I)$ for prescribed line $L$ and interval $I$. It is not clear how to do this for arbitrarily small intervals because for general networks there can be many arbitrarily close critical points, but we will only need to do so for certain "nice" $I$ as suggested by Theorem 4.8.

To that end, first note that it is straightforward to form the quantities $b_L(I)$ for intervals $I$ entirely contained within linear pieces of $F|_L$; we formalize this in Algorithm 1.

It remains to demonstrate how to construct $\nabla_L(I)$. Intuitively one can accomplish this via "finite differencing," i.e. the gradient of a piecewise linear function $F$ at a point $x$ can be computed from queries by computing $\frac{F(x+\delta) - F(x)}{\delta}$ for several sufficiently small perturbations $\delta \in \mathbb{R}^d$ and solving the linear system.

With *a priori* precision estimates, we can similarly implement a gradient oracle, as formalized in Algorithm 2 and Lemma 4.9.

---

**Algorithm 1:** GETBIAS($L, I$)

**Input:** Line $L \subset \mathbb{R}^d$, interval $I = [a, b] \subset \mathbb{R}$
**Output:** $b_L(I)$ if $I$ is entirely contained within a linear piece of $F|_L$

1  $t_{\mathsf{mid}} \leftarrow$ midpoint of $I$.
2  $y_0 \leftarrow F|_L(t_{\mathsf{mid}})$.
3  $s \leftarrow \frac{F|_L(b) - F|_L(a)}{b - a}$.
4  **return** $y_0 - s \cdot t_{\mathsf{mid}}$

---

**Algorithm 2:** GETGRADIENT($x, \alpha$)

**Input:** $x \in \mathbb{R}^d$, $\alpha > 0$ for which (6) holds
**Output:** $\nabla F(x) \in \mathbb{R}^d$

1  **for** $j \in [d]$ **do**
2     Sample random unit vector $z_j \in \mathbb{S}^{d-1}$.
3     $v_j \leftarrow (F(x + \alpha z_j) - F(x))/\alpha$.
4  Let $w$ be the solution to the linear system $\{\langle w, z_j \rangle = v_j\}_{j \in [d]}$.
5  **return** $w$

---

**Lemma 4.9.** *For any $\alpha > 0$ and any $x \in \mathbb{R}^d$ for which*

$$|\langle w_i, x \rangle - b_i| \geq \alpha \|w_i\| \ \ \forall \, i \in [k], \tag{6}$$

GETGRADIENT*($x, \alpha$) makes $d$ queries to $F$ and outputs $\nabla F(x)$.*

In order to use GETGRADIENT to construct the vectors $\nabla_L(I)$, we require estimates for $\alpha$ in Lemma 4.9. In the following lemma we show that with high probability over the randomness of $L$, if an interval $I$ completely lies within a linear piece of $F|_L$, then we can bound how small we must take $\alpha$ to query the gradient of $F$ at the midpoint of that interval. As is clear from the proof of Theorem 4.8 in the supplementary material, these are the only intervals that we need gradient and bias information from.

**Lemma 4.10.** *Let $L$ be a Gaussian line. With probability at least $1 - \delta$ over the randomness of $L$, the following holds: in the partition $[-\tau, \tau] = I_1 \cup \cdots \cup I_m$ in Theorem 4.8, for any $I_\ell$ which entirely lies within a linear piece of $F|_L$,* GETGRADIENT*($t_{\mathsf{mid}}, \alpha$) correctly outputs $\nabla_L(I_\ell)$, where $x_{\mathsf{mid}}$ is the midpoint of the interval $\mathcal{I}_\ell \subset L$ that corresponds to interval $I_\ell \subset \mathbb{R}$ and $\alpha = \frac{\delta \cdot r}{4k\sqrt{d} + O(k\sqrt{\log(k/\delta)})}$ (where $r$ is defined in (3)).*

Putting these ingredients together, we obtain the following algorithm, GETNEURONS for producing a collection of neurons that can be used to approximate $F$.

We prove correctness of GETNEURONS in the following lemma:

**Lemma 4.11.** *For any $\epsilon, \delta > 0$,* GETNEURONS*($\epsilon, \delta$) makes $\mathrm{poly}(k, d, R, B, 1/\epsilon, \log(1/\delta))$ queries and outputs a list $\mathcal{S}$ of pairs $(w, b)$ for which there exist $\{\pm 1\}$-valued coefficients $\{a_{w,b}\}_{(w,b) \in \mathcal{S}}$ as well as a vector $w^*$ and a scalar $b^*$ such that*

$$\left\| F - \langle w^*, \cdot \rangle - b^* - \sum_{(w,b) \in \mathcal{S}} a_{w,b} \cdot \sigma(\langle w, \cdot \rangle - b) \right\| \leq \epsilon.$$

### 4.4   Linear Regression Over ReLU Features

It remains to show how to combine the neurons produced by GETNEURONS to obtain a good approximation to $F$. As Theorem 4.8 already ensures that some linear combination of them suffices, we can simply draw many samples $(x, F(x))$ for $x \sim \mathcal{N}(0, \mathrm{Id})$, form the feature vectors computed by the neurons output by GETNEURONS, and run linear regression on these feature vectors.

Formally, let $\mathcal{S}$ denote the set of pairs $(w, b)$ guaranteed by Theorem 4.8. We will denote the $w$'s by $\{\widehat{w}_j\}$ and the $b$'s by $\{\widehat{b}_j\}$. Consider the following distribution over feature vectors computed by the neurons in $\mathcal{S}$:

---
**Algorithm 3:** GETNEURONS($\epsilon, \delta$)
---
**Input:** Accuracy $\epsilon > 0$, confidence $\delta > 0$
**Output:** List $\mathcal{S}$ of pairs $(w, b)$ (see Theorem 4.8 for guarantee)

1   $\mathcal{S} \leftarrow \emptyset$.
2   Sample Gaussian line $L$.
3   $\Delta \leftarrow (\epsilon / \mathfrak{P}_{k,R,B,\log(1/\delta)})^{9/2}$.           `// Theorem 4.8`
4   $\alpha \leftarrow \frac{\delta \cdot r}{4k\sqrt{d} + O(k\sqrt{\log(k/\delta)})}$.           `// Lemma 4.10`
5   Define $r, \tau$ according to (3), (4).
6   Partition $[-\tau, \tau]$ into disjoint intervals $I_1, \ldots, I_m$ of length $r$.
7   **for** *all* $j \in [m]$ **do**
8      $x_j \leftarrow$ midpoint of the interval $\mathcal{I}_j \subset L$ that corresponds to $I_j \subset \mathbb{R}$.
9      $\widehat{\nabla}_L(I_j) \leftarrow$ GETGRADIENT$(x_j, \alpha)$.
10     $\widehat{b}_L(I_j) \leftarrow$ GETBIAS$(L, I_j)$.
11   **for** *all pairs of distinct* $i, j \in [m]$ **do**
12     $(v_j, b_j) \leftarrow (\widehat{\nabla}_L(I_i) - \widehat{\nabla}_L(I_j), \widehat{b}_L(I_i) - \widehat{b}_L(I_j))$.
13     **if** $(v_j, b_j)$ *satisfies the bounds in* (5) **then**
14       Add $(v_j, b_j)$ to $\mathcal{S}$.

15   **return** $\mathcal{S}$.
---

**Definition 4.12.** *Let $\mathcal{D}'$ denote the distribution over $\mathbb{R}^{|\mathcal{S}|+d+1} \times \mathbb{R}$ of pairs $(z, y)$ given by sampling $x \sim \mathcal{N}(0, Id)$ and forming the vector $z$ whose entries consist of all $\sigma(\langle \widehat{w}_j, x \rangle - \widehat{b}_j)$ as well as the entries of $x$ and the entry 1, and taking $y$ to be $F(x)$ for the ground truth network $F$ defined in (2).*

*We will also need to define a truncated version of $\mathcal{D}'$: let $\mathcal{D}$ denote $\mathcal{D}'$ conditioned on the norm of the $|\mathcal{S}|+1$ to $|\mathcal{S}|+d$-th coordinates having norm at most $M \triangleq \sqrt{d} + O(\sqrt{\log(1/\delta)})$, which happens with probability at least $1 - \delta$ over $\mathcal{D}'$.*

Our algorithm will be to sample sufficiently many pairs $(z, y)$ from $\mathcal{D}'$ (by querying $F$ on random Gaussian inputs) and run ordinary least squares. This is outlined in LEARNFROMQUERIES below.

---
**Algorithm 4:** LEARNFROMQUERIES($\epsilon, \delta$)
---
**Input:** Accuracy $\epsilon > 0$, confidence $\delta > 0$
**Output:** One hidden-layer network $\widetilde{F} : \mathbb{R}^d \to \mathbb{R}$ for which $\|F - \widetilde{F}\| \le O(\epsilon)$

1   $\mathcal{S} = \{(\widehat{w}_j, \widehat{b}_j)\} \leftarrow$ GETNEURONS$(\epsilon, \delta)$.
2   Draw samples $(z_1, y_1), \ldots, (z_n, y_n)$ from $\mathcal{D}$         `// Definition 4.12`
3   Let $\widetilde{v}$ be the solution to the least-squares problem (7). Let $\widetilde{b}$ denote the last entry of $\widetilde{v}$, and let $\widetilde{w}$ denote the vector given by the $d$ entries of $\widetilde{v}$ prior to the last.
4   Form the network $\widetilde{F}(x) \triangleq \sum_j \widetilde{v}_j \sigma(\langle \widehat{w}_j, x \rangle - \widehat{b}_j) + \langle \widetilde{w}, \cdot \rangle - \widetilde{b}$.
5   **return** $F$.
---

Formally, we show:

**Theorem 4.13.** *Let $\mathcal{S}$ denote the list of pairs $(\widehat{w}_j, \widehat{b}_j)$ output by GETNEURONS$(\epsilon, \delta)$. Sample $(z_1, y_1), \ldots, (z_n, y_n)$ from $\mathcal{D}$ for $n = \text{poly}(k, R, B, 1/\epsilon, d, \log(1/\delta))$. With probability at least $1 - O(\delta)$ over the randomness of GETNEURONS and the samples, the following holds. Define*

$$\widetilde{v} \triangleq \arg \min_{\|v\| \le W} \sum_{i=1}^{n} (\langle v, z_i \rangle - y_i)^2, \text{ for } W \triangleq \sqrt{\tau/r} + k(R + B), \tag{7}$$

*let $\widetilde{b}$ denote the last entry of $\widetilde{v}$, and let $\widetilde{w}$ denote the vector given by the $d$ entries of $\widetilde{v}$ prior to the last. Then the one hidden-layer network $\widetilde{F}(x) \triangleq \sum_j \widetilde{v}_j \sigma(\langle \widehat{w}_j, x \rangle - \widehat{b}_j) + \langle \widetilde{w}, \cdot \rangle - \widetilde{b}$ satisfies $\|F - \widetilde{F}\| \le O(\epsilon)$.*

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
