# 1 Supplementary Materials Roadmap

In this supplementary material, we provide "full versions" of Sections 2-4 from the main submission, corresponding to Sections A-C in the sequel. While it is easiest for the reader to read section 1 of the main submission and directly jump to Sections A-C below, we will match the names of the definitions/lemmas/theorems/etc. from the main submission with those in the supplement.

In Section A we give notation and technical preliminaries that will be useful in our subsequent proofs. In Section B we prove our key structural results on approximating linear combinations of many similar neurons by small networks. Finally, in Section C, we give our full algorithm for learning one-hidden-layer networks from queries.

## A    Preliminaries

**Notation.**   Given vectors $u, v$, let $\angle(u,v) \triangleq \arccos\left(\frac{\langle u,v \rangle}{\|u\|\|v\|}\right)$. Let $e_j$ denote the $j$-th standard basis vector in $\mathbb{R}^d$. We will occasionally denote the standard Gaussian measure on $\mathbb{R}$ by $d\gamma(x)$. Given a function $h$ which is square-integrable with respect to the Gaussian measure, we will use $\|h\|$ to denote $\mathbb{E}_{x \sim \mathcal{N}(0,\mathrm{Id})}[h(x)^2]^{1/2}$. Given a collection of indices $S \subseteq \mathbb{Z}$, we say that $i, j \in S$ are *neighboring* if there does not exist $i < \ell < j$ for which $\ell \in S$.

The following elementary fact will be useful:

**Fact A.1.** $|\sin(x+y)| = |\sin(x)\cos(y) + \sin(y)\cos(x)| \le |\sin(x)| + |\sin(y)|$ *for any* $x, y \in \mathbb{R}$

### A.1    Neural Networks, Restrictions, and Critical Points

**Definition 2.1.** *A neuron is a pair* $(v, b)$ *where* $v \in \mathbb{R}^d$ *and* $b \in \mathbb{R}$; *it corresponds to the function* $x \mapsto \sigma(\langle v,x \rangle - b)$, *which we sometimes denote by* $\sigma(\langle v, \cdot \rangle - b)$.

As mentioned in the overview, we will be taking random restrictions of the underlying network $F$, for which we use the following notation:

**Definition 2.2.** *Given a line* $L \subset \mathbb{R}^d$ *parametrized by* $L = \{x_0 + t \cdot v\}_{t \in \mathbb{R}}$, *and a function* $F : \mathbb{R}^d \to \mathbb{R}$, *define the* restriction of $F$ to $L$ *by* $F|_L(t) \triangleq F(x_0 + t \cdot v)$.

**Definition 2.3.** *Given a line* $L \subset \mathbb{R}^d$ *and a restriction* $F|_L$ *of a piecewise linear function* $F : \mathbb{R}^d \to \mathbb{R}$ *to that line, the* critical points *of* $F|_L$ *are the points* $t \in \mathbb{R}$ *at which the slope of* $F|_L$ *changes.*

### A.2    Concentration and Anti-Concentration

We will need the following standard tail bounds and anti-concentration bounds:

**Fact 2.4** (Concentration of norm of Gaussian vector). *Given Gaussian vector* $h \sim \mathcal{N}(0, \Sigma)$, $\Pr\left[\|h\| \ge O(\|\Sigma^{1/2}\|_{\mathsf{op}}(\sqrt{r} + \sqrt{\log(1/\delta)}))\right] \le \delta$, *where* $r$ *is the rank of* $\Sigma$.

**Fact 2.5** (Uniform bound on entries of Gaussian vector). *For covariance matrix* $\Sigma \in \mathbb{R}^{m \times m}$, *given* $h \sim \mathcal{N}(0, \Sigma)$ *we have that* $|h_i| \le O\left(\sqrt{\Sigma_{i,i}}\sqrt{\log(m/\delta)}\right)$ *for all* $i \in [m]$ *with probability at least* $1 - \delta$.

*Proof.* For every $i \in [m]$, $h_i \sim \mathcal{N}(0, \Sigma_{i,i})$, so $|h_i| \le O(\Sigma_{i,i}^{1/2}\sqrt{\log(m/\delta)})$ with probability at least $1 - \delta/m$, from which the claim follows by a union bound and the fact that the largest diagonal entry of a psd matrix is the largest entry of that matrix. $\square$

**Fact A.2** (Carbery-Wright [CW01]). *There is an absolute constant* $C > 0$ *such that for any* $\nu > 0$ *and quadratic polynomial* $p : \mathbb{R}^d \to \mathbb{R}$, $\Pr_{g \sim \mathcal{N}(0,\mathrm{Id})}[|p(g)| \le \nu \cdot \mathbb{V}[p(g)]^{1/2}] \le C\sqrt{\nu}$.

**Lemma 2.6** (Anti-concentration of norm of Gaussian vector). *There is an absolute constant* $C > 0$ *such that given any Gaussian vector* $h \sim \mathcal{N}(\mu, \Sigma)$, $\Pr\left[\|h\| \ge \sqrt{\nu}\|\Sigma\|_F^{1/2}\right] \ge 1 - C\sqrt{\nu}$.

*Proof.* Define the polynomial $p(g) \triangleq (g + \mu)^\top \Sigma (g + \mu)$. Note that for $g \sim \mathcal{N}(0, \text{Id})$, $p(g)$ is distributed as $\|h\|^2$ for $h \sim \mathcal{N}(\mu, \Sigma)$. We have $\mathbb{E}_{g \sim \mathcal{N}(0, \text{Id})}[p(g)] = \text{Tr}(\Sigma) + \mu^\top \Sigma \mu$, so

$$
\begin{aligned}
\mathbb{V}[p(g)] &= \mathbb{E}[(g^\top \Sigma g + 2g^\top \Sigma \mu - \text{Tr}(\Sigma))^2] \\
&= \mathbb{E}[(g^\top \Sigma g)^2] + \mathbb{E}[(2g^\top \Sigma \mu - \text{Tr}(\Sigma))^2] + 2\,\mathbb{E}[(g^\top \Sigma g)(2g^\top \Sigma \mu - \text{Tr}(\Sigma))] \\
&= \left(2\,\text{Tr}(\Sigma^2) + \text{Tr}(\Sigma)^2\right) + \left(4\,\text{Tr}(\Sigma \mu \mu^\top \Sigma) + \text{Tr}(\Sigma)^2\right) - 2\,\text{Tr}(\Sigma)^2 \\
&= 2\langle \Sigma^2, \text{Id} + 2\mu\mu^\top \rangle \geq \|\Sigma\|_F^2,
\end{aligned}
$$

so by Fact A.2 we conclude that $\Pr[p(g) \leq \nu\|\Sigma\|_F] \leq C\sqrt{\nu}$. $\qquad\square$

**Lemma 2.7** (Anti-concentration for random unit vectors). *For random* $v \in \mathbb{S}^{d-1}$,

$$
\Pr\left[|v_1| < \frac{\delta}{2\sqrt{d} + O(\sqrt{\log(1/\delta)})}\right] \leq \delta.
$$

*Proof.* For $g \sim \mathcal{N}(0, \text{Id})$, $g/\|g\|$ is identical in distribution to $v$. $\|g\| \leq \sqrt{d} + O(\sqrt{\log(1/\delta)})$ with probability at least $1 - \delta/2$ for absolute constant $c > 0$, and furthermore $\Pr_{\gamma \sim \mathcal{N}(0,1)}[|g| > t] \geq 1 - t$ for any $t > 0$, from which the claim follows by a union bound. $\qquad\square$

# B ReLU Networks with Cancellations

In the following section we prove several general results about approximating one hidden-layer networks with many "similar" neurons by much smaller networks.

## B.1 Stability Bounds for ReLUs

The main result of this subsection will be the following stability bound for (non-homogeneous) ReLUs with the same bias.

**Lemma B.1.** *Fix any* $\Delta < 1$. *For orthogonal* $v, v' \in \mathbb{R}^d$ *for which* $\|v - v'\| \leq \Delta\|v\|$, *and* $b \in \mathbb{R}$, *we have*

$$
\mathbb{E}[(\sigma(\langle v, x \rangle - b) - \sigma(\langle v', x \rangle - b))^2] \leq O\left(\Delta^{2/5}\|v\|^2\right)
$$

To prove this, we will need to collect some standard facts about stability of homogeneous ReLUs and affine threshold functions, given in Fact B.2, Lemma B.3, Lemma B.4, and Lemma B.5.

The following formula is standard [CS09]:

**Fact B.2.** $\mathbb{E}[\sigma(\langle v, x \rangle)\sigma(\langle v', x \rangle)] = \frac{1}{2\pi}\|v\|\|v'\|\left(\sin\angle(v, v') + (\pi - \angle(v, v'))\cos\angle(v, v')\right)$. *For* $\langle v, v' \rangle \geq 0$, *note that this is at least* $\frac{1}{6}\|v\|\|v'\| + \frac{1}{3}\langle v, v' \rangle$.

As a consequence, we obtain the following stability result for homogeneous ReLUs:

**Lemma B.3.** *For any* $v, v' \in \mathbb{R}^d$ *for which* $\langle v, v' \rangle \geq 0$, *we have*

$$
\mathbb{E}[(\sigma(\langle v, x \rangle) - \sigma(\langle v', x \rangle))^2] \leq \frac{1}{2}\|v - v'\|^2 + \frac{2}{3}\|v\|\|v'\|(1 - \cos\angle(v, v'))
$$

*Proof.* We can expand the expectation and apply Fact B.2 to get

$$
\begin{aligned}
\mathbb{E}[(\sigma(\langle v, x \rangle) - \sigma(\langle v', x \rangle))^2] &= \mathbb{E}[\sigma(\langle v, x \rangle)^2] + \mathbb{E}[\sigma(\langle v', x \rangle)^2] - 2\,\mathbb{E}[\sigma(\langle v, x \rangle)\sigma(\langle v', x \rangle)] \\
&\leq \frac{1}{2}\|v\|^2 + \frac{1}{2}\|v'\|^2 - 2\left(\frac{1}{6}\|v\|\|v'\| + \frac{1}{3}\langle v, v' \rangle\right) \\
&= \frac{1}{2}\|v - v'\|^2 + \frac{2}{3}(\|v\|\|v'\| - \langle v, v' \rangle) \\
&= \frac{1}{2}\|v - v'\|^2 + \frac{2}{3}\|v\|\|v'\|(1 - \cos\angle(v, v'))
\end{aligned}
$$

as claimed. $\qquad\square$

We will also need the following stability result for affine linear thresholds.

**Lemma B.4** (Lemma 5.7 in [CM20])**.** *Given $v, v' \in \mathbb{R}^d$ and $b \in \mathbb{R}$,*

$$\Pr[\langle v, x \rangle > b \wedge \langle v', x \rangle \le b] \le O(\|v - v'\|/b).$$

**Lemma B.5.** *For any $v \in \mathbb{R}^d$ and $b \le b'$,*

$$\mathbb{E}\big[(\sigma(\langle v, x \rangle - b) - \sigma(\langle v, x \rangle - b'))^2\big] \le (b' - b)^2$$

*Proof.* Note that $\langle v, x \rangle \sim \mathcal{N}(0, \|v\|^2)$, so it suffices to show that for the univariate function $f(z) \triangleq \sigma(z - b) - \sigma(z - b')$, $\mathbb{E}_{z \sim \mathcal{N}(0, \|v\|^2)}[f(z)^2] \le (b' - b)^2$. Observe that $f(z) = b' - b$ for $z > b'$, $f(z) = 0$ for $z < b$, and $f(z) = z - b$ for $z \in [b, b']$. In particular, $|f(z)| \le b' - b$, from which the claim follows. $\square$

The following basic lemma giving $L_2$ bounds for Lipschitz functions which are bounded with high probability will be useful throughout.

**Lemma B.6.** *Let $\epsilon(x) : \mathbb{R}^d \to \mathbb{R}_{\ge 0}$ be any square-integrable function with respect to the Gaussian measure. If $f : \mathbb{R}^d \to \mathbb{R}$ is an $L$-Lipschitz continuous piecewise linear function and satisfies $\Pr_{x \sim \mathcal{N}(0, Id)}[|f(x)| \le \epsilon(x)] \ge 1 - \zeta$ and $|f(0)| \le M$, then $\mathbb{E}_{x \sim \mathcal{N}(0, Id)}[f(x)^2] \le 2\zeta M^2 + L^2 \zeta^{1/2}(d^2 + 2d) + \mathbb{E}[\epsilon(x)^4]^{1/2}$.*

*Proof.* Because $f$ is $L$-Lipschitz, $f(x)^2 \le (M + L\|x\|)^2 \le 2M^2 + L^2\|x\|^2$. Then

$$\begin{aligned}
\mathbb{E}[f(x)^2] &\le \mathbb{E}[f(x)^2 \mathbb{1}[f(x) > \epsilon(x)]] + \mathbb{E}[\epsilon(x)^2 \mathbb{1}[f(x) \le \epsilon(x)]] \\
&\le 2\zeta M^2 + L^2 \mathbb{E}[\|x\|^2 \mathbb{1}[f(x) > \rho\|x\|]] + \mathbb{E}[\epsilon(x)^4]^{1/2}(1 - \zeta)^{1/2} \\
&\le 2\zeta M^2 + L^2 \zeta^{1/2} \mathbb{E}[\|x\|^4] + \mathbb{E}[\epsilon(x)^4]^{1/2}(1 - \zeta)^{1/2} \\
&= 2\zeta M^2 + 3L^2 \zeta^{1/2} d^2 + \mathbb{E}[\epsilon(x)^4]^{1/2},
\end{aligned}$$

as claimed. $\square$

Putting all of these ingredients together, we can now complete the proof of the main Lemma B.1 of this subsection.

*Proof.* Suppose $b \ge \Delta^{1/5}\|v\|$. By Lemma B.4, $\mathrm{sgn}(\langle v, x \rangle - b) \neq \mathrm{sgn}(\langle v', x \rangle - b)$ with probability at most $O(\Delta\|v\|/b)$. So with probability at least $1 - O(\Delta\|v\|/b)$, the function $(\sigma(\langle v, x \rangle - b) - \sigma(\langle v', x \rangle - b)$ is at most $\langle v - v', x \rangle \le \Delta\|v\|\|x\|$. Furthermore, this function is $L$-Lipschitz for $L = \|v\| + \|v\|' \le O(\|v\|)$. By Lemma B.6 applied to the projection of $f$ to the two-dimensional subspace spanned by $v, v'$,

$$\mathbb{E}[(\sigma(\langle v, x \rangle - b) - \sigma(\langle v', x \rangle - b))^2] \lesssim \|v\|^2 \left(\sqrt{\Delta\|v\|/b} + \Delta^2\right) \lesssim \Delta^{2/5}\|v\|^2.$$

Now suppose $b < \Delta^{1/5}\|v\|$. Then $\|\sigma(\langle v, \cdot \rangle - b) - \sigma(\langle v, \cdot \rangle)\|^2 \le \Delta^{2/5}\|v\|^2$ and $\|\sigma(\langle v', \cdot \rangle - b) - \sigma(\langle v', \cdot \rangle)\|^2 \le \Delta^{2/5}\|v'\|^2$. By triangle inequality, it suffices to bound $\|\sigma(\langle v, \cdot \rangle) - \sigma(\langle v', \cdot \rangle)\|^2$. By Lemma B.3, we have

$$\|\sigma(\langle v, \cdot \rangle) - \sigma(\langle v', \cdot \rangle)\|^2 \lesssim \Delta^2\|v\|^2 + \|v\|^2 \cdot (1 - \cos \angle(v, v')) \lesssim \Delta^2\|v\|^2,$$

where the last step follows by the fact that $\|v - v'\| \le \Delta\|v\|$ implies that $\cos \angle(v, v') \ge \sqrt{1 - \Delta^2} \ge 1 - \Delta^2$. $\square$

## B.2 $(\Delta, \alpha)$-Closeness of Neurons

We now formalize a notion of geodesic closeness between two neurons and record some useful properties. This notion is motivated by Lemma 4.4 in Section C.1 where we study the critical points of random restrictions of one hidden-layer networks.

**Definition 3.1.** *Given $v, v' \in \mathbb{R}^d$ and $b, b' \in \mathbb{R}$, we say that $(v, b)$ and $(v', b')$ are $(\Delta, \alpha)$-close if the following two conditions are satisfied:*

*1. $|\sin \angle(v, v')| \le \Delta$*

100       2. $\|bv' - b'v\| \le \alpha\|v\|\|v'\|$.

101 *Note that this is a measure of angular closeness between* $(v, b), (v', b') \in \mathbb{R}^{d+1}$. *For instance, if*
102 $(v, b) = (\lambda v^*, \lambda b^*)$ *and* $(v', b') = (\lambda' v^*, \lambda' b^*)$ *for some* $(v^*, b^*)$, *then* $(v, b)$ *and* $(v', b')$ *are* $(0, 0)$-
103 *close.*

104 We first collect some elementary consequences of closeness. The following intuitively says that if
105 we scale two $(\Delta, \alpha)$-neurons to have similar norm, then their biases will be close.

106 **Lemma 3.2.** *If* $(v, b)$ *and* $(v', b')$ *are* $(\Delta, \alpha)$-*close, and* $v = \gamma v' + v^\perp$ *for* $v^\perp$ *orthogonal to* $v'$, *then*
107 $|\gamma b' - b| \le \alpha\|v\|$.

108 *Proof.* We know that $\|bv' - b'v\| \le \alpha\|v\|\|v'\|$. The left-hand side of this is $\|(b - \gamma b')v' - b'v^\perp\| \ge$
109 $|b - \gamma b'|\|v'\|$, where the inequality follows from orthogonality of $v, v'$. Therefore, $|\gamma b' - b| \le \alpha\|v\|$
110 as claimed. □

111 Note that when two neurons are $(\Delta, \alpha)$-close, their weight vectors are either extremely correlated or
112 extremely anti-correlated. In fact, given a collection of neurons that are all pairwise close, they will
113 exhibit the following "polarization" effect.

114 **Lemma 3.3.** *Suppose* $\Delta < \sqrt{2}/2$. *If* $(v_1, b_1), \ldots (v_k, b_k)$ *are all pairwise* $(\Delta, \alpha)$-*close for some*
115 $\alpha > 0$, *then there is a partition* $[k] = S_1 \sqcup S_2$ *for which* $\langle v_i, v_j \rangle \ge 0$ *for any* $i \in S_1, j \in S_1$ *or*
116 $i \in S_2, j \in S_2$, *and for which* $\langle v_i, v_j \rangle < 0$ *for any* $i \in S_1, j \in S_2$ *or* $i \in S_2, j \in S_1$.

117 *Proof.* Let $S_1$ be the set of $i \in [k]$ for which $\langle v_i, v_1 \rangle \ge 0$, and let $S_2$ be the remaining indices. First
118 consider any $i, j \in S_1$ and note that $\angle(v_i, v_j) \le \angle(v_i, v_1) + \angle(v_j, v_1) \le 2 \arcsin \Delta$, and because
119 $\langle v_i, v_1 \rangle, \langle v_j, v_1 \rangle \ge 0$, this is less than $\pi/4$ for $\Delta < \sqrt{2}/2$. By the same reasoning, we can show that
120 for any $i, j \in S_2$, $\angle(v_i, v_j) < \pi/2$ if $\Delta < \sqrt{2}/2$. Finally, consider $i \in S_1$ and $j \in S_2$. We have
121 $\angle(v_i, v_j) \ge \angle(v_j, v_1) - \angle(v_i, v_1)$. If $\Delta < \sqrt{2}/2$, then $\angle(v_j, v_1) > 3\pi/4$ while $\angle(v_i, v_1) < \pi/4$,
122 concluding the proof. □

123 In the rest of the paper we will take $\Delta$ to be small, so Lemma 3.3 will always apply. As such, it will
124 be useful to define the following terminology:

125 **Definition 3.4.** *Given* $(v_1, b_1), \ldots, (v_k, b_k)$ *which are all pairwise-close, we will call the partition*
126 $S_1 \sqcup S_2$ *given in Lemma 3.3 the* orientation *induced by* $\{(v_i, b_i)\}$.

127 We note that $(\Delta, \alpha)$-closeness satisfies triangle inequality.

128 **Lemma 3.5.** *If* $(v_1, b_1)$ *and* $(v_2, b_2)$ *are* $(\Delta, \alpha)$-*close, and* $(v_2, b_2)$ *and* $(v_3, b_3)$ *are* $(\Delta', \alpha')$-*close,*
129 *then* $(v_1, b_1)$ *and* $(v_3, b_3)$ *are* $(\Delta + \Delta', 2\alpha + 2\alpha')$-*close.*

130 *Proof.* As $\angle(v_1, v_3) \le \angle(v_1, v_2) + \angle(v_2, v_3)$, it is clear from Fact A.1 that $|\sin \angle(v_1, v_3)| \le \Delta + \Delta'$.

131 Now write the orthogonal decompositions $v_1 = \gamma_1 v_2 + v_1^\perp$ and $v_3 = \gamma_3 v_2 + v_3^\perp$, noting that
132 $\gamma_1\|v_2\| \le \|v_1\|, \gamma_3\|v_2\| \le \|v_3\|$. We can write

$$b_1 v_3 - b_3 v_1 = (b_1 \gamma_3 - b_3 \gamma_1)v_2 + (b_1 v_3^\perp - b_3 v_1^\perp). \tag{1}$$

133 We will handle these two terms separately. First note that $(\Delta, \alpha)$-closeness of $(v_1, b_1), (v_2, b_2)$ and
134 Lemma 3.2 imply $|b_2 \gamma_1 - b_1| \le \alpha\|v_1\|$, so in particular $|b_2 \gamma_1 \gamma_3 - b_1 \gamma_3| \le \alpha\gamma_3\|v_1\|$. Similarly,
135 $|b_2 \gamma_1 \gamma_3 - b_3 \gamma_1| \le \alpha'\gamma_1\|v_3\|$. This allows us to conclude by triangle inequality that

$$|b_1\gamma_3 - b_3\gamma_1| \cdot \|v_2\| \le \alpha\gamma_1\|v_3\| + \alpha'\gamma_3\|v_1\|)\|v_2\| \le (\alpha + \alpha')\|v_1\|\|v_3\|. \tag{2}$$

136 It remains to handle the second term on the right-hand side of (1). Note that Lemma 3.2 also tells us
137 that

$$\|b_1 v_3^\perp - b_3 v_1^\perp\| \le \|b_2\gamma_1 v_3^\perp - b_1 v_3^\perp\| + \|b_2\gamma_3 v_1^\perp - b_3 v_1^\perp\| \le \alpha\|v_1\|\|v_3^\perp\| + \alpha'\|v_3\|\|v_1^\perp\| \le (\alpha + \alpha')\|v_1\|\|v_3\|,$$
$$\tag{3}$$

138 so by (1), (2), and (3), $\|b_1 v_3 - b_3 v_1\| \le 2(\alpha + \alpha')\|v_1\|\|v_3\|$. □

## B.3 Merging Neurons

In this section we begin to apply the tools we have developed in the preceding sections to show our main results about approximating neural networks with many close neurons by smaller networks. The goal of this subsection is to prove that a one hidden-layer network where all neurons are $(\Delta, \alpha)$-close to some neuron can be approximated by at most two neurons:

**Lemma 3.6.** *Given $F(x) = \sum_{i=1}^{k} s_i \sigma(\langle w_i, x \rangle - b_i)$ for $s_i \in \{\pm 1\}$ and $(v^*, b^*) \in \mathbb{R}^d \times \mathbb{R}$ for which $(w_i, b_i)$ is $(\Delta, \alpha)$-close to $(v^*, b^*)$ for all $i \in [k]$, there exist coefficients $a^+, a^- \in \mathbb{R}$ for which*

$$\mathbb{E}_{x \sim \mathcal{N}(0, Id)} \left[ \left( F(x) - a^+ \sigma(\langle v^*, x \rangle - b^*) - a^- \sigma(\langle -v^*, x \rangle + b^*) \right)^2 \right] \leq O(k^2 (\Delta^{2/5} + \alpha^2)) \|v^*\|^2.$$

(4)

*Furthermore, we have that*

$$|a^+| \|v^*\|, |a^-| \|v^*\| \leq \sum_i \|w_i\| \qquad and \qquad |a^+ b^*|, |a^- b^*| \leq \alpha \sum_i \|w_i\| + \sum_i |b_i|. \quad (5)$$

Our starting point for showing this is the following lemma which states that given *two* close neurons whose weight vectors are correlated, we can merge them into a single neuron while incurring small square loss.

**Lemma B.7.** *Let $0 < \Delta \leq 1$. For $v_1, v_2, v \in \mathbb{R}^d$, suppose we have $v_1 = \gamma_1 v + v_1^\perp$ and $v_2 = \gamma_2 v + v_2^\perp$ for $1 \geq \gamma_1 \geq \gamma_2 \geq 0$ and $v_1^\perp, v_2^\perp$ orthogonal to $v$. Suppose additionally that $(v_1, b_1)$ and $(v_2, b_2)$ are both $(\Delta, \alpha)$-close to $(v, b)$. For $s \in \{\pm 1\}$, we have that*

$$\mathbb{E}_{x \sim \mathcal{N}(0, Id)} \left[ \left( \sigma(\langle v_1, x \rangle - b_1) + s \sigma(\langle v_2, x \rangle - b_2) - (\gamma_1 + s\gamma_2) \sigma(\langle v, x \rangle - b) \right)^2 \right] \leq O\left( \Delta^{2/5} + \alpha^2 \right) \|v\|^2$$

*Proof.* For $i = 1, 2$, because $|\sin \angle(v_i, v)| \leq \Delta$, we find $\|v_i^\perp\| \leq \Delta \|v_i\| \leq O(\Delta \|v\|)$ for $\Delta$ sufficiently small. From Lemma B.1 we have $\|\sigma(\langle v_i, \cdot \rangle - b_i) - \sigma(\langle \gamma_i v, \cdot \rangle - b_i)\| \leq O(\Delta^{1/5} \|v\|)$. Note that

$$(\gamma_i b - b_i) \|v\|^2 = b \langle v, v_i \rangle - b_i \|v\|^2 \leq \|v\| \|b v_i - b_i v\| \leq \alpha \|v\|^2 \|v_i\|,$$

i.e. $\gamma_i b - b_i \leq \alpha \|v_i\|$. So by Lemma B.5, $\|\sigma(\langle \gamma_i v, \cdot \rangle - b_i) - \sigma(\langle \gamma_i v, \cdot \rangle - \gamma_i b)\| \leq \alpha \|v_i\|$. The lemma follows by triangle inequality and the fact that $\|v_i\| \leq \|v\| \sqrt{1 + \Delta^2} \leq 2 \|v\|$. $\qquad \square$

Lemma B.7 suggests the following binary operation.

**Definition B.8.** *Fix a vector $v^* \in \mathbb{R}^d$. Consider the set of all triples $(s, v, b)$ for which $s \in \{\pm 1\}$, $b \in \mathbb{R}$, and $v$ satisfies $0 \leq \langle v, v^* \rangle \leq \|v^*\|^2$. Define the binary operator $\odot_{v^*}$ as follows. Suppose $v_1 = \gamma_1 v + v_1^\perp$ and $v_2 = \gamma_2 v + v_2^\perp$ as in Lemma B.7, and define $\gamma = |s_1 \gamma_1 + s_2 \gamma_2|$. Then*

$$(s_1, v_1, b_1) \odot_{v^*} (s_2, v_2, b_2) = (s_i, \gamma v, \gamma b) \text{ for } i = \arg\max_j \gamma_j$$

*Note that $s_i$ corresponds to the sign of $s_1 \gamma_1 + s_2 \gamma_2$, and $s_i \gamma = s_1 \gamma_1 + s_2 \gamma_2$.*

In this notation we can restate Lemma B.7 as follows:

**Lemma B.9.** *For $v_1, v_2, b_1, b_2, v$, satisfying the conditions of Lemma B.7, if we define the tuple $(s', v', b')$ by $(s', v', b') = (s_1, v_1, b_1) \odot_v (s_2, v_2, b_2)$ we have that*

$$\mathbb{E}_{x \sim \mathcal{N}(0, Id)} \left[ \left( s_1 \sigma(\langle v_1, x \rangle - b_1) + s_2 \sigma(\langle v_2, x \rangle - b_2) - s' \sigma(\langle v', x \rangle - b') \right)^2 \right] \leq O\left( \Delta^{2/5} + \alpha^2 \right) \|v\|^2$$

It will be useful to record some basic properties of this binary operation:

**Fact B.10.** *$\odot_{v^*}$ is associative and commutative. Moreover, if $(s_1, v_1, b_1) \odot_{v^*} \cdots \odot_{v^*} (s_m, v_m, b_m) = (s, \gamma v, \gamma b)$ for $s$ given by the sign of $\sum_i s_i \gamma_i$, where $v_i = \gamma_i v^* + v_i^\perp$ for $v_i^\perp$ orthogonal to $v^*$, then $s$ is the sign of $\sum s_i \gamma_i$, and $s\gamma = \sum s_i \gamma_i$.*

*Proof.* That $\odot_{v^*}$ is commutative is evident from the definition. For associativity, consider $(s_1, v_1, b_1), (s_2, v_2, b_2), (s_3, v_3, b_3)$. Recall that if $(s_1, v_1, b_1) \odot_{v^*} (s_2, v_2, b_2) = (s_i, \gamma_{12} v^*, \gamma_{12} b)$

for $\gamma_{12} = |s_1\gamma_1 + s_2\gamma_2|$, then $s_i$ corresponds to the sign of $s_1\gamma_1 + s_2\gamma_2$, so $s_i\gamma = s_1\gamma_1 + s_2\gamma_2$. We conclude that

$$((s_1, v_1, b_1) \odot_{v^*} (s_2, v_2, b_2)) \odot_{v^*} (s_3, v_3, b_3) = (s_i, \gamma_{12}v^*, \gamma_{12}b) \odot_{v^*} (s_3, v_3, b_3) = (s_{i'}, \gamma_{123}v^*, \gamma_{123}b)$$

for $\gamma_{123} = |s_1\gamma_1 + s_2\gamma_2 + s_3\gamma_3|$ and $s_{i'}$ corresponding to the sign of $s_1\gamma_1 + s_2\gamma_2 + s_3\gamma_3$. It is therefore evident that $\odot_{v^*}$ is associative. The last part of the claim follows by induction. $\qquad\square$

We show that merging many neurons which are all $(\Delta, \alpha)$ close to some given neuron $\sigma(\langle v^*, \cdot \rangle - b^*)$ results in a neuron which is also close to $\sigma(\langle v^*, \cdot \rangle - b^*)$.

**Lemma B.11.** *Let $m > 1$. Given $v_1, \ldots, v_m, v^* \in \mathbb{R}^d$ and $b_1, \ldots, b_m, b^*$ for which every $(v_i, b_i)$ is $(\Delta, \alpha)$-close to $(v^*, b^*)$ and satisfies $\langle v_i, v^* \rangle \geq 0$, we have that for*

$$(s, v, b) \triangleq (s_1, v_1, b_1) \odot_{v^*} \cdots \odot_{v^*} (s_m, v_m, b_m),$$

*$(v, b)$ is $(0,0)$-close to $(v^*, b^*)$ and satisfies $\langle v, v^* \rangle \geq 0$. Furthermore, $\|v\| \leq \sum_i \|v_i\|$ and $|b| \leq \alpha \sum_i \|v_i\| + \sum_i |b_i|$.*

*Proof.* Suppose first that $m = 2$. As usual, let $\Pi_{v^*} v_i = \gamma_i v^*$. Recall that $v = \gamma v^*$ and $b = \gamma b^*$ for $\gamma = |s_1\gamma_1 + s_2\gamma_2|$. As a result, we clearly have that $\langle v, v^* \rangle \geq 0$. Furthermore,

$$\|bv^* - b^*v\| = \|\gamma b^* v^* - \gamma b^* v^*\| = 0.$$

The first part of the claim then follows by induction. For the norm bound, note that $\|v\| = |\sum_i \gamma_i| \cdot \|v^*\| \leq \sum_i \|v_i\|$. For the bound on $|b|$, recall from Lemma 3.2 that for every $i$, $\|\gamma_i b^* - b_i\| \leq \alpha\|v_i\|$. So $|b| = |\sum_i \gamma_i b^*| \leq \sum_i (|b_i| + \alpha\|v_i\|)$ as claimed. $\qquad\square$

Putting everything from this subsection together, we are now ready to prove Lemma 3.6:

*Proof of Lemma 3.6.* Denote $\odot_{v^*}$ by $\odot$. Let $S^+$ denote the set of $i \in [k]$ for which $\langle v^*, v_i \rangle \geq 0$, and let $S^-$ denote the remaining indices $i \in [k]$. Define $F^+(x) \triangleq \sum_{i \in S^+} \sigma(\langle w_i, x \rangle - b_i)$ and $F^-(x) \triangleq \sum_{i \in S^-} \sigma(\langle w_i, x \rangle - b_i)$. By Lemma B.7, Lemma B.11, and triangle inequality, we have that for $(s^+, w^+, b^+) \triangleq \bigodot_{i \in S^+} (s_i, w_i, b_i)$ and $(s^-, w^-, b^-) \triangleq \bigodot_{i \in S^-} (s_i, w_i, b_i)$,

$$\|F^+ - s^+\sigma(\langle w^+, \cdot \rangle - b^+)\|^2, \|F^- - s^-\sigma(\langle w^-, \cdot \rangle - b^-)\|^2 \leq O(k^2(\Delta^{2/5} + \alpha^2))\|v^*\|^2.$$

Recalling that $(w^+, b^+) = (\gamma^+ v^*, \gamma^+ b^*)$ and $(w^-, b^-) = (\gamma^- v^*, \gamma^- b^*)$, we conclude the proof of (4) with one more application of triangle inequality. For the bounds in (5), we simply apply the last part of Lemma B.11. $\qquad\square$

## B.4 Constructing a Close Neuron

Note that Lemma 3.6 requires the existence of a neuron $(v^*, b^*)$ which is close to all neurons $\{(v_i, b_i)\}$. In our algorithm, we will not have access to $(v^*, b^*)$ but rather to some linear combination of the neurons $\{(v_i, b_i)\}$. We first show that provided this linear combination is not too small in norm, it will also be close to all the neurons $\{(v_i, b_i)\}$.

**Lemma 3.7.** *Suppose we have vectors $v_1, \ldots, v_m, v^* \in \mathbb{R}^d$, biases $b_1, \ldots, b_m, b^* \in \mathbb{R}$ for which every $(v_i, b_i)$ is $(\Delta, \alpha)$-close to $(v^*, b^*)$. Then for any $s_1, \ldots, s_m \in \{\pm 1\}$, if we define $v \triangleq \sum_{i=1}^m s_i v_i$ and $b \triangleq \sum_{i=1}^m s_i b_i$, then $(v, b)$ is $(\Delta m, \alpha \sum_i \|v_i\|/\|v\|)$-close to $(v^*, b^*)$.*

*Proof.* Note that $\angle(\sum_i s_i v_i, v) \leq \sum_i \angle(v_i, v)$. By Fact A.1, we have that $\sin\angle(\sum_i s_i v_i, v) \leq \Delta m$.

The lemma then follows from noting that

$$\|b^* v - b v^*\| = \left\|\sum_i s_i(b_i v^* - b^* v_i)\right\| \leq \alpha\|v^*\| \cdot \sum_i \|v_i\| = \alpha\|v\|\|v^*\| \cdot \sum_i \|v_i\|/\|v\|.$$

$\qquad\square$

## B.5 A Corner Case

This presents an issue: what if the linear combination of neurons that we get access to in our eventual algorithm has small norm, in which case Lemma 3.7 is not helpful? It turns out this linear combination takes a very specific form (see the vector in (6)), and we argue in this section that if it is indeed small, then the underlying network we are trying to approximate will be close to a linear function! The main result of this subsection is to show:

**Lemma 3.8.** *Suppose* $(v_1, b_1), \ldots, (v_m, b_m)$ *are pairwise* $(\Delta, \alpha)$-*close, and let* $[m] = S_1 \sqcup S_2$ *denote the orientation induced by them (see Definition 3.4). If signs* $s_1, \ldots, s_m \in \{\pm 1\}$ *satisfy*

$$\|\sum_{i \in S_1} s_i v_i - \sum_{i \in S_2} s_i v_i\| \leq (\Delta R)^{2/9}, \tag{6}$$

*then for the network* $F(x) \triangleq \sum_i s_i \sigma(\langle v_i, x \rangle - b_i)$, *there exists an affine linear function* $\ell(x) : \mathbb{R}^d \to \mathbb{R}$ *for which*

$$\mathbb{E}_{x \sim \mathcal{N}(0, Id)}\left[(F(x) - \ell(x))^2\right] \leq \text{poly}(k, R, B) \cdot (\alpha^{1/2} + \Delta^{2/9}) \tag{7}$$

*where* $B \triangleq \max_i \|b_i\|$ *and* $R \triangleq \max_i \|v_i\|$, *and* $\ell \triangleq \langle w^*, \cdot \rangle - b^*$ *satisfying*

$$\|w^*\| \leq \sum_i \|v_i\| \qquad and \qquad |b^*| \leq \sum_i \|b_i\|. \tag{8}$$

Before proceeding to the proof, we will need the following stability result for affine linear threshold functions with possibly different thresholds.

**Lemma B.12.** *Suppose* $(v, b)$ *and* $(v', b')$ *are* $(\Delta, \alpha)$-*close and* $\|v\| \geq \|v'\|$. *If* $\langle v, v' \rangle \geq 0$ *then*

$$\Pr[\langle v, x \rangle > b \wedge \langle v', x \rangle < b'] \leq O\left(\alpha + \sqrt{\Delta \|v\| / \|v'\|}\right). \tag{9}$$

*Otherwise, if* $\langle v, v' \rangle < 0$, *then*

$$\Pr[\langle v, x \rangle > b \wedge \langle v', x \rangle > b'] \leq O\left(\alpha + \sqrt{\Delta \|v\| / \|v'\|}\right).$$

*Proof.* Clearly it suffices to prove (9). Suppose $\|v'\| \leq \|v\|$ and write $v' = \gamma v + v^\perp$ for $v^\perp$ orthogonal to $v$. Note that $\|v^\perp\| \leq \Delta \|v'\|$ and that $\|v^\perp\| \leq \gamma \|v\| \cdot \tan \angle(v, v') \leq O(\gamma \Delta \|v\|)$ for $\Delta$ sufficiently small.

Note that

$$\Pr[\text{sgn}(\langle v', x \rangle - \gamma b) \neq \text{sgn}(\langle v', x \rangle - b')] \leq \Pr_{g \sim \mathcal{N}(0, \|v'\|^2)}[g \in [\gamma b \wedge b', \gamma b \vee b']] \leq \frac{|b' - \gamma b|}{2\|v'\|}. \tag{10}$$

Because $\|bv' - b'v\| = \|(b\gamma - b')v + bv^\perp\| \leq \alpha \|v\| \|v'\|$, we have that $|b\gamma - b'| \leq \alpha \|v'\|$. We conclude that $\Pr[\text{sgn}(\langle v', x \rangle - \gamma b) \neq \text{sgn}(\langle v', x \rangle - b')] \leq \alpha/2$.

So by a union bound it suffices to bound $\Pr[\langle \gamma v, x \rangle > \gamma b \wedge \langle v', x \rangle < \gamma b]$. By Lemma B.4, this is at most $\frac{\|v' - \gamma v\|}{\gamma b} = \frac{1}{\gamma b} \|v^\perp\| \leq O(\frac{\Delta \|v\|}{b})$.

We can also bound this in a different way. By a similar calculation to (10), we have $\Pr[\text{sgn}(\langle \gamma v, x \rangle - \gamma b) \neq \text{sgn}(\langle \gamma v, x \rangle)] \leq \frac{b}{2\|v\|}$ and $\Pr[\text{sgn}(\langle v', x \rangle - b) \neq \text{sgn}(\langle v', x \rangle)] \leq \frac{b}{2\|v'\|}$. And by Sheppard's formula, $\Pr[\text{sgn}(\langle v, x \rangle) \neq \text{sgn}(\langle v', x \rangle)] \leq \frac{\angle(v, v')}{\pi} \leq O(\Delta)$ for $\Delta$ sufficiently small.

We conclude that

$$\Pr[\langle \gamma v, x \rangle > \gamma b \wedge \langle v', x \rangle < \gamma b] \lesssim \frac{\Delta \|v\|}{b} \wedge \left(\frac{b}{\|v'\|} + \Delta\right) \lesssim \sqrt{\Delta \|v\| / \|v'\|},$$

from which the claim follows. $\square$

We can now prove Lemma 3.8.

*Proof of Lemma 3.8.* Define $\omega \triangleq \|\sum_{i \in S_1} s_i v_i - \sum_{i \in S_2} s_i v_i\|$. Let $S_0 \subseteq [m]$ denote the set of $i$ for which $\|v_i\| \leq (\Delta R)^{1/9}$. For $i \in S_0$, note that by Lipschitz-ness of the ReLU function,

$$\|\sigma(\langle v_i, \cdot \rangle - b_i) - \sigma(-b_i)\|^2 \leq \|\langle v_i, \cdot \rangle\|^2 = \|v_i\|^2 \leq \Delta^{2/9} R^{2/9}.$$

So by triangle inequality it suffices to show that $\sum_{i \notin S_0} s_i \sigma(\langle v_i, x \rangle - b_i)$ is well-approximated by some affine linear function. We will thus assume without loss of generality that $S_0 = \emptyset$.

By Lemma B.12 and a union bound over all pairs $i, j \in [m]$, we have that with probability at least $1 - O(m^2 \alpha + m^2 \Delta^{4/9} R^{4/9})$ over $x \sim \mathcal{N}(0, \text{Id})$, $\text{sgn}(\langle v_i, x \rangle - b_i) = \text{sgn}(\langle v_j, x \rangle - b_j)$ is the same for all $i, j \in S_1$ and for all $i, j \in S_2$, and $\text{sgn}(\langle v_i, x \rangle - b_i) \neq \text{sgn}(\langle v_j, x \rangle - b_j)$ for all $i \in S_1, j \in S_2$. Let $\mathbb{1}[x \in \mathcal{E}]$ denote the indicator for this event. In other words, with high probability all of the neurons in $S_1$ are activated and none in $S_2$ are, or vice versa; denote these two events by $\mathcal{E}_1$ and $\mathcal{E}_2$ respectively.

For $j = 1, 2$, note that when $x \in \mathcal{E}_j$, $F(x) = \left\langle \sum_{i \in S_j} s_i v_i, x \right\rangle - \sum_{i \in S_j} s_i b_i$. Define $\ell(x) = \left\langle \sum_{i \in S_1} s_i v_i, x \right\rangle - \sum_{i \in S_1} s_i b_i$. Obviously when $x \in S_1$, $F(x) = \ell(x)$. To handle $x \in S_2$, we need to bound $\delta \triangleq \left| \sum_{i \in S_1} s_i b_i - \sum_{i \in S_2} s_i b_i \right|$. Let $(v, b) = (v_1, b_1)$ and note that because $(v_i, b_i)$ is $(\Delta, \alpha)$-close to $(v, b)$ for all $i$,

$$\alpha \|v\| \sum_i \|v_i\| \geq \left\| \left( \sum_{i \in S_1} s_i b_i - \sum_{i \in S_2} s_i b_i \right) v - b \left( \sum_{i \in S_1} s_i v_i - \sum_{i \in S_2} s_i v_i \right) \right\| \geq \delta \|v\| - |b| \omega.$$

In particular, $\delta \leq \alpha \sum_i \|v_i\| + |b| \omega / \|v\| \leq \alpha R + B \omega / (\Delta R)^{1/9}$.

We would like to apply Lemma B.6 to $F(x) - \ell(x)$ (projected to the span of $\{v_i\}$). In that lemma, we can take $\epsilon(x) \leq \left| \langle \sum_{i \in S_1} s_i v_i - \sum_{i \in S_2} s_i v_i, x \rangle \right| + \delta$, for which we have $\mathbb{E}[\epsilon(x)^4]^{1/2} \leq O(\delta^2 + \omega^2)$. Additionally we can naively bound $F(0) - \ell(0) \leq 2 \sum_i |b_i|$ and therefore take $M$ in that lemma to be $2 \sum_i |b_i| \leq 2mB$. In addition, we can take $\zeta = O(m^2 \alpha + m^2 \Delta^{4/9} R^{4/9})$, $L = 2mR$, and $d =,$.

We conclude that

$$\mathbb{E}[(F(x) - \ell(x))^2] = \mathbb{E}[(F(x) - \ell(x))^2 \mathbb{1}[x \in \mathcal{E}_2]] + \mathbb{E}[(F(x) - \ell(x))^2 \mathbb{1}[x \notin \mathcal{E}]]$$
$$\lesssim (m\alpha^{1/2} + m\Delta^{2/9} R^{2/9}) \cdot (m^2 B^2 + m^4 R^2) + \alpha^2 R^2 + B^2 \omega^2 / (\Delta R)^{2/9} + \omega^2.$$

Recalling that we paid an additional $m^2 (\Delta R)^{2/9}$ in square loss in reducing to the case where $S_0 = \emptyset$, we obtain the desired bound in (7). The bounds in (8) follow immediately from the definition of $\ell$ above. $\qquad \square$

## B.6 Putting Everything Together

Putting Lemmas 3.6, 3.7, and 3.8 together, we conclude that networks whose hidden units are pairwise $(\Delta, \alpha)$-close can either be approximated by a particular size-two network, or by *some* affine linear function:

**Lemma 3.9.** *Suppose $(v_1, b_1), \ldots, (v_k, b_k)$ are pairwise $(\Delta, \alpha)$-close, and let $[k] = S_1 \sqcup S_2$ denote the orientation induced by them (see Definition 3.4). Define $B \triangleq \max_i \|b_i\|$ and $R \triangleq \max_i \|v_i\|$. Let $s_1, \ldots, s_m \in \{\pm 1\}$.*

*Define $F(x) = \sum_i s_i \sigma(\langle v_i, x \rangle - b_i)$, $v^* = \sum_{i \in S_1} s_i v_i - \sum_{i \in S_2} s_i v_i$, and $b^* + = \sum_{i \in S_1} s_i b_i - \sum_{i \in S_2} s_i b_i$. At least one of the following holds:*

1. *There is an affine linear function $\ell : \mathbb{R}^d \to \mathbb{R}$ for which $\|F - \ell\|^2 \leq \text{poly}(k, R, B) \cdot (\alpha^{1/2} + \Delta^{2/9})$.*

2. *There exist coefficients $a^+, a^- \in \mathbb{R}$ for which $G(x) \triangleq a^+ \sigma(\langle v^*, x \rangle - b^*) - a^- \sigma(\langle -v^*, x \rangle + b^*)$ satisfies $\|F - G\|^2 \leq \text{poly}(k, R, B) \cdot (\Delta^{2/5} + \alpha^2 \Delta^{-4/9})$.*

*Proof.* By assumption, every $(v_i, b_i)$ is $(\Delta, \alpha)$-close to $(v_1, b_1)$. By Lemma 3.7 we get that for $(v^*, b^*)$ defined in the lemma statement, $(v_1, b_1)$ is $(\Delta k, \alpha mR/\|v^*\|)$-close to $(v^*, b^*)$.

If $\|v^*\| \geq (\Delta R)^{2/9}$, then we conclude that $(v_1, b_1)$ is $(\Delta k, \alpha m \Delta^{-2/9} R^{7/9})$-close to $(v^*, b^*)$, and by Lemma 3.6 we find that there is a choice of $a^+, a^-$ for which the function $G$ defined in the lemma statement satisfies $\|F - G\|^2 \leq O(k^4 R^2 (\Delta^{2/5} k^{2/5} + \alpha^2 m^2 \Delta^{-4/9} R^{14/9}))$ (note that we used $\|v^*\| \leq \sum_i \|v_i\| \leq kR$).

If $\|v^*\| \leq (\Delta R)^{2/9}$, then by Lemma 3.8 we find that there is an affine linear $\ell$ for which $\|F - \ell\|^2 \leq$ poly$(k, R, B) \cdot (\alpha^{1/2} + \Delta^{2/9})$. $\qquad\square$

# C  Learning One Hidden Layer

In this section we give our algorithm for learning neural networks from queries. Throughout, we will suppose we have black-box query access to some unknown one-hidden layer neural network

$$F(x) \triangleq \sum_{i=1}^{k} s_i \sigma(\langle w_i, x \rangle - b_i), \qquad (11)$$

where $s_i \in \{\pm 1\}$, $w_i \in \mathbb{R}^d$, $b_i \in \mathbb{R}$. Define the quantities $R \triangleq \max_i \|w_i\|$ and $B \triangleq \max_i |b_i|$; our bounds will be polynomial in these quantities, among others.

In Section C.1, we give bounds on the separation among critical points of random restrictions of $F$. In Section C.2 we prove our main *existence theorem* showing that by carefully searching along a random restriction of $F$, we are able to recover a collection of neurons that can be combined to approximate $F$. In Section C.3 we show how to implement certain key steps in GETNEURONS involving querying the gradient and bias of $F$ at certain points. Finally, in Section C.4 we show to find an appropriate combination of these neurons.

## C.1  Critical Points of One-Hidden Layer Networks

In this section, we compute the critical points of restrictions of $F$ and argue that they are far apart along *random restrictions* unless if the corresponding neurons were close to begin with (in the sense of Definition 3.1).

First, we formalize the notion of a random restriction:

**Definition 4.1.** *A* Gaussian line $L$ *is a random line in $\mathbb{R}^d$ formed as follows: sample $x_0 \sim \mathcal{N}(0, Id)$ and Haar-random $v \in \mathbb{S}^{d-1}$ and form the line $L \triangleq \{x_0 + t \cdot v\}_{t \in \mathbb{R}}$.*

Here we compute the critical points along a restriction of $F$.

**Proposition 4.2.** *Given a line $L = \{x_0 + t \cdot v\}_{t \in \mathbb{R}}$, the restriction $F|_L(t) \triangleq F(x_0 + t \cdot v)$ is given by*

$$F|_L(t) = \sum_{i=1}^{k} s_i \sigma \left( \langle w_i, x_0 \rangle - b_i + t \langle w_i, v \rangle \right).$$

*This function has $k$ critical points, namely $t = -\frac{\langle w_i, x_0 \rangle - b_i}{\langle w_i, v \rangle}$ for every $i \in [k]$.*

*Proof.* The critical points of $F|_L$ are precisely the points $t$ at which a neuron changes sign. So the crticial point associated to the $i$-th neuron is the $t$ for which $\langle w_i, x_0 \rangle - b_i + t \langle w_i, v \rangle = 0$, from which the claim follows. $\qquad\square$

We can show that these critical points are not too large, unless the norm of the corresponding weight vector is small. The reason for the latter caveat is that, e.g., if one took the one-dimensional neuron $\sigma(\epsilon z - b)$ for $b$ fixed and $\epsilon \to 0$, the $z$ at which it changes sign tends to $\infty$).

**Lemma 4.3.** *With probability at least $1 - \delta$ over the randomness of Gaussian line L, we have that $|t_i| \lesssim \frac{k(\sqrt{d} + \sqrt{\log(1/\delta)})}{\delta \|w_i\|} + k\left(\sqrt{d} + \sqrt{\log(1/\delta)}\right) \sqrt{\log(k/\delta)}$ for every critical point $t_i$ of $F|_L$.*

*Proof.* By Lemma 2.7, with probability $1 - \delta$ we have that $|\langle w_i, v \rangle| \gtrsim \frac{\delta \|w_i\|}{k(\sqrt{d} + \sqrt{\log(1/\delta)})}$ for all $i \in [k]$. Also note that $|\langle w_i, x_0 \rangle| \leq \|w_i\| \cdot \sqrt{\log(k/\delta)}$ for all $i \in [k]$ by Fact 2.5. By Proposition 4.2,

the critical point corresponding to the $i$-th hidden unit satisfies

$$|t| = \left| \frac{\langle w_i, x_0 \rangle - b_i}{\langle w_i, v \rangle} \right| \lesssim \frac{k(\sqrt{d} + \sqrt{\log(1/\delta)})}{\delta \|w_i\|} \left( B + \|w_i\| \sqrt{\log(k/\delta)} \right)$$

$$\leq \frac{k(\sqrt{d} + \sqrt{\log(1/\delta)})}{\delta \|w_i\|} + k \left( \sqrt{d} + \sqrt{\log(1/\delta)} \right) \sqrt{\log(k/\delta)}.$$

$\square$

Fix a separation parameter $\Delta > 0$ which we will tune in the sequel. We show that along Gaussian lines $L$, $F|_L$'s critical points are well-separated except for those corresponding to neurons which are $(\Delta, \alpha)$-close.

**Lemma 4.4.** *There is an absolute constant $c > 0$ for which the following holds. Given Gaussian line $L$, with probability at least $1 - \delta$ we have: for any pair of $i, j$ for which $(w_i, b_i)$ and $(w_j, b_j)$ are not $(\Delta, c\Delta\sqrt{\log(k/\delta)})$-close, the corresponding critical points are at least $\Omega \left( \frac{\Delta \delta^2}{k^4 \left( \sqrt{d} + \sqrt{\log(k/\delta)} \right)} \right)$- apart.*

*Proof.* For every $i \in [k]$, let $t_i \triangleq -\frac{\langle w_i, x_0 \rangle - b_i}{\langle w_i, v \rangle}$ denote the location of the critical point corresponding to neuron $i$. For any $i, j \in [k]$,

$$|t_j - t_i| = \left| \frac{\langle w_j, v \rangle(\langle w_i, x_0 \rangle - b_i) - \langle w_i, v \rangle(\langle w_j, x_0 \rangle - b_j)}{\langle w_i, v \rangle \langle w_j, v \rangle} \right|$$

$$\geq \frac{|\langle ((\langle w_i, x_0 \rangle w_j - \langle w_j, x_0 \rangle w_i) - (b_i w_j - b_j w_i)), v \rangle|}{\|w_i\| \|w_j\|} \triangleq |\langle z_{ij}, v \rangle|.$$

Note that $(\langle w_i, x_0 \rangle w_j - \langle w_j, x_0 \rangle w_i) - (b_i w_j - b_j w_i)$ is distributed as $\mathcal{N}(\mu, \Sigma)$ for $\mu = -b_i w_j + b_j w_i$ and $\Sigma^{1/2} = w_j w_i^\top - w_i w_j^\top$. One can verify that

$$\|\Sigma\|_F^{1/2} = 2^{1/4} \left( \|w_i\|^2 \|w_j\|^2 - \langle w_i, w_j \rangle^2 \right)^{1/2} = 2^{1/4} \|w_i\| \|w_j\| |\sin \angle(w_i, w_j)|$$

For the first part of the lemma, suppose $|\sin \angle(w_i, w_j)| \geq \Delta$ so that $\|\Sigma\|_F^{1/2} \geq \Omega(\Delta \|w_i\| \|w_j\|)$. Then by Lemma 2.6 we conclude that $\|z_{ij}\| \geq \Omega(\Delta \delta / k^2)$ with probability at least $1 - \delta/k^2$. Recall that $v$ is a random unit vector drawn independently of $x_0$, so the lemma follows by applying Lemma 2.7 and a union bound over all pairs $i, j$.

On the other hand, suppose $|\sin \angle(w_i, w_j)| \leq \Delta$ but $\|\mu\| \geq c\Delta \sqrt{\log(k/\delta)} \|w_i\| \|w_j\|$ for $c > 0$ sufficiently large. Note that $\Sigma$ has rank 2, so by Fact 2.4, the norm of a sample from $\mathcal{N}(0, \Sigma)$ has norm at most $O(\|\Sigma^{1/2}\|_{\text{op}}(\sqrt{2} + \sqrt{\log(k/\delta)})) = O(\Delta \|w_i\| \|w_j\| \sqrt{\log(k/\delta)})$ with probability at least $1 - \delta/k^2$. So if we take $c$ large enough that this is at least $\Omega \left( \frac{\Delta \delta^2}{k^4 \sqrt{d}} \right)$ less than $c\Delta \|w_i\| \|w_j\| \sqrt{\log(k/\delta)}$, we conclude that $\|z_{ij}\| \geq \Omega(\Delta \delta / k^2)$ with probability at least $1 - \delta/k^2$. $\square$

## C.2 Line Search and Existence Theorem

At a high level, our algorithm works by searching along $F|_L$, partitioning $L$ into small intervals, and computing differences between the gradients/biases of $F$ at the midpoints of these intervals. The primary structural result we must show is that there exists enough information in this set of differences to reconstruct $F$ up to small error.

As we will be working with partitions of lines, it will be convenient to define the following notation:

**Definition 4.5.** *Given line $L \subset \mathbb{R}^d$ and finite interval $I \subseteq \mathbb{R}$ corresponding to a segment $\mathcal{I} \subset L$, let $\nabla_L(I)$ denote the gradient of $F$ at the midpoint of $\mathcal{I}$. For $t_{\text{mid}} \in \mathbb{R}$ the midpoint of $I$, define $b_L(I) \triangleq F|_L(t_{\text{mid}}) - (F|_L)'(t_{\text{mid}}) \cdot t_{\text{mid}}$. Intuitively, this is the "y-intercept" of the linear piece of $F|_L$ that contains $t_{\text{mid}}$. When $L$ is clear from context, we will drop subscripts and denote these objects by $\nabla(I)$ and $b(I)$.*

**Definition 4.6.** *Given line $L \subset \mathbb{R}^d$ and length $s > 0$, let $\{t_i\}$ denote the critical points of $F|_L$, and let $G_L^+(s) \subseteq [k]$ (resp. $G_L^-(s)$) denote the set of indices $a \leq i \leq b$ for which $t_{i+1} - t_i \geq s$ (resp. $t_i - t_{i-1} \geq s$). Let $G_L^*(r) \triangleq G_L^+(r) \cap G_L^-(r)$.*

The following observation motivates Definition 4.6:

**Observation 4.7.** *Given line $L \subset \mathbb{R}^d$, let $\{t_i\}$ denote the critical points of $F|_L$. Let $I_1, \ldots, I_m$ be a partition of some interval $I$ into pieces of length $r$, and for $t_i \in I$ let $\ell(i)$ denote the index of the interval containing $I$.*

*Then for any $i \in G_L^+(2r)$ (resp. $i \in G_L^-(2r)$), $I_{\ell(i)+1}$ is entirely contained within $[t_i, t_{i+1}]$ (resp. $I_{\ell(i)-1}$ is entirely contained within $[t_{i-1}, t_i]$). In particular, $I_{\ell(i)-1}$ and $I_{\ell(i)+1}$ are linear pieces of $F|_L$.*

The following is the main result of this section. At a high level, it says if we partition a random line in $\mathbb{R}^d$ into sufficiently small intervals and can compute the gradient of $F$ at the midpoint of each interval, then we can produce a collection of neurons which can be used to approximate $F$.

**Theorem 4.8.** *For any $\epsilon, \delta > 0$, define*

$$r \triangleq O\left(\frac{\Delta \delta^2}{k^4 \left(\sqrt{d} + \sqrt{\log(k/\delta)}\right)}\right) \tag{12}$$

$$\tau \triangleq kr + \Theta\left(\frac{k(\sqrt{d} + \sqrt{\log(1/\delta)})}{\delta \|w_i\|} + k\left(\sqrt{d} + \sqrt{\log(1/\delta)}\right)\sqrt{\log(k/\delta)}\right). \tag{13}$$

*Partition the interval $[-\tau, \tau]$ into intervals $I_1, \ldots, I_m$ of length $r$.*

*Let $L$ be a Gaussian line, and let $\mathcal{S}$ denote the set of all $m(m-1)$ pairs $(w, b)$ obtained by taking distinct $i, j \in [k]$ and forming $(\nabla_L(I_i) - \nabla_L(I_j), b_L(I_i) - b_L(I_j))$. There exist $\{\pm 1\}$-valued coefficients $\{a_{w,b}\}_{(w,b)\in\mathcal{S}}$, vector $w^*$, and $b^* \in \mathbb{R}$ for which*

$$\left\|F - \sum_{(w,b)\in\mathcal{S}} a_{w,b} \cdot \sigma(\langle w, \cdot \rangle - b) - \langle w^*, \cdot \rangle - b^*\right\| \leq \epsilon + \mathfrak{P}_{k,R,B,\log(1/\delta)} \cdot \Delta^{2/9}.$$

*for $\mathfrak{P}_{k,R,B,\log(1/\delta)}$ some absolute constant that is polynomially large in $k, R, B, \log(1/\delta)$. Furthermore, we have that*

$$\|a_{w,b} \cdot w\| \leq kR \qquad and \qquad |a_{w,b} \cdot b| \leq c\Delta k^2 R\sqrt{\log(k/\delta)} + kB \tag{14}$$

$$\|w^*\| \leq kR \qquad and \qquad |b^*| \leq kB \tag{15}$$

*Proof.* Condition on the outcomes of Lemma 4.4 and Lemma 4.3 holding for $L$. Let $t_1, \ldots, t_k$ denote the critical points associated to neurons $w_1, \ldots, w_k$, and for convenience we assume without loss of generality that $t_1 \leq \cdots \leq t_k$. Let $a, b \in [k]$ denote the indices for which $|t_i| \leq \tau$ for $i \in [a, b]$. By Lemma 4.3 and the definition of $\tau$, we have that for $i \notin [a, b]$, $\|w_i\| \leq \epsilon/k$.

By Lipschitzness of the ReLU function,

$$\left\|\sum_{i\notin[a,b]} s_i\sigma(\langle w_i, \cdot \rangle - b_i) - \sum_{i\notin[a,b]} s_i\sigma(-b_i)\right\| \leq \sum_{i\notin[a,b]} \|\sigma(\langle w_i, \cdot \rangle - b_i) - \sigma(-b_i)\|$$

$$\leq \sum_{i\notin[a,b]} \|w_i\| \leq (b - a + 1)\epsilon/k. \tag{16}$$

Next, we handle the critical points $i \in [a, b]$. Given critical point $t_i$, let $\ell(i) \in [m]$ denote the index for which $t_i \in I_{\ell(i)}$. For convenience, denote $G_L^+(2r), G_L^-(2r), G_L^*(2r)$ by $G^+, G^-, G^*$. By Observation 4.7, we know that for $i \in G^*$, the linear piece of $F|_L$ immediately preceding critical point $t_i$ contains $I_{\ell(i)-1}$, and the one immediately proceeding $t_i$ contains $I_{\ell(i)+1}$. Therefore,

$\nabla(I_{\ell(i)+1}) - \nabla(I_{\ell(i)-1})$ and $b(I_{\ell(i)-1}) - b(I_{\ell(i)+1})$ are equal to $w_i$ and $b_i$ up to a sign, so $\mathcal{S}$ must contain the neurons $(w_i, b_i)$ and $(-w_i, -b_i)$.

Now consider any neighboring $i_1 < i_2$ in $G^+ \Delta G^-$ for which $i_2 - i_1 > 1$; note that the latter condition implies that $i_1 \in G^- \backslash G^+$ and $i_2 \in G^+ \backslash G^-$, or else we would have a violation of the fact that $i_1$ and $i_2$ are neighboring. Furthermore, because $i_1, i_2$ are neighboring, for all $i_1 \le i \le i_2$ we have that $t_{i+1} - t_i \le 2r$. By taking $\Delta$ in (the contrapositive of) Lemma 4.4 to be $\Delta \cdot k$, we conclude that for any $i_1 \le i < j \le i_2$, $(w_i, b_i)$ and $(w_j, b_j)$ are $(\Delta k, c\Delta k \sqrt{\log(k/\delta)})$-close for all such $i$.

Let $\{i_1, \ldots, i_2\} = S_1 \sqcup S_2$ denote the orientation induced by $(w_{i_1}, b_{i_1}), \ldots (w_{i_2}, b_{i_2})$. We would like to apply Lemma 3.9 to the subnetwork $\widetilde{F}(x) \triangleq \sum_{j=i_1}^{i_2} s_j \sigma(\langle w_j, x \rangle - b_j)$. By another application of Observation 4.7, we know that $\nabla(I_{\ell(i_2)}) - \nabla(I_{\ell(i_1)})$ and $b(I_{\ell(i_1)}) - b(I_{\ell(i_2)})$ are, up to a common sign, precisely the vector $v^*$ and bias $b^*$ defined in Lemma 3.9, so we conclude that either there exists a network $G$ consisting of neurons $\sigma(\langle v^*, x \rangle - b^*)$ and $\sigma(\langle -v^*, x \rangle + b^*)$ for which $\|\widetilde{F} - G\|^2 \le \text{poly}(k, R, B) \cdot (\Delta^{2/5} k^{2/5} + c^2 \Delta^{14/9} k^2 \log(k/\delta)) \le \text{poly}(k, R, B) \Delta^{2/5} \log(1/\delta)$, or there is an affine linear function $\ell$ for which $\|\widetilde{F} - \ell\|^2 \le \text{poly}(k, R, B) \cdot (c^{1/2} \Delta^{1/2} \log(1/\delta)^{1/2} + \Delta^{2/9}) \le \text{poly}(k, R, B) \cdot \Delta^{2/9} \log(1/\delta)^{1/2}$. Furthermore, the bounds in (14) and (15) follow from (5) in Lemma 3.6 (for $\alpha = c\Delta k \sqrt{\log(k/\delta)}$) and (8) in Lemma 3.8 respectively.

We have accounted for all critical points, except in the case where the smallest index $a'$ in $G^-$ is not $a$, or the largest index $b'$ in $G^+$ is not $b$. In the former (resp. latter) case, note that $t_a \le \cdots \le t_{a'-1} \le -\tau + kr$, (resp. $t_b \ge \cdots \ge t_{b'+1} \ge \tau - kr$), so by Lemma 4.3, this implies that $\|w_{a'-1}\|, \ldots, \|w_a\| \le \epsilon/k$ (resp. $\|w_{b'+1}\|, \ldots, \|w_b\| \le \epsilon/k$). By Lipschitzness of the ReLU function, we can approximate these neurons by constants at a total cost of at most $(a' - a + b - b')\epsilon/k$ in $L_2$ using the same reasoning as (16). $\qquad\square$

## C.3   Gradient and Bias Oracles

It remains to implement oracles to compute $b_L(I)$ and $\nabla_L(I)$ for prescribed line $L$ and interval $I$. It is not clear how to do this for arbitrarily small intervals because for general networks there can be many arbitrarily close critical points, but we will only need to do so for certain "nice" $I$ as suggested by Theorem 4.8.

To that end, first note that it is straightforward to form the quantities $b_L(I)$ for intervals $I$ entirely contained within linear pieces of $F|_L$; we formalize this in Algorithm 1.

---

**Algorithm 1:** GETBIAS$(L, I)$

**Input:** Line $L \subset \mathbb{R}^d$, interval $I = [a, b] \subset \mathbb{R}$
**Output:** $b_L(I)$ if $I$ is entirely contained within a linear piece of $F|_L$
1  $t_{\text{mid}} \leftarrow$ midpoint of $I$.
2  $y_0 \leftarrow F|_L(t_{\text{mid}})$.
3  $s \leftarrow \frac{F|_L(b) - F|_L(a)}{b-a}$.
4  **return** $y_0 - s \cdot t_{\text{mid}}$

---

It remains to demonstrate how to construct $\nabla_L(I)$. Intuitively one can accomplish this via "finite differencing," i.e. the gradient of a piecewise linear function $F$ at a point $x$ can be computed from queries by computing $\frac{F(x+\delta) - F(x)}{\delta}$ several sufficiently small perturbations $\delta \in \mathbb{R}^d$ and solving the linear system.

With *a priori* precision estimates, we can similarly implement a gradient oracle, as formalized in Algorithm 2 and Lemma 4.9.

**Lemma 4.9.** *For any $\alpha > 0$ and any $x \in \mathbb{R}^d$ for which*

$$|\langle w_i, x \rangle - b_i| \ge \alpha \|w_i\| \ \ \forall i \in [k], \tag{17}$$

GETGRADIENT$(x, \alpha)$ *makes $d$ queries to $F$ and outputs $\nabla F(x)$.*

*Proof.* For any $z \in \mathbb{S}^{d-1}$, note that

$$\langle w_i, x + \alpha z \rangle - b_i = (\langle w_i, x \rangle - b_i) + \alpha \langle w_i, z \rangle,$$

---

**Algorithm 2:** GETGRADIENT$(x, \alpha)$

---

**Input:** $x \in \mathbb{R}^d$, $\alpha > 0$ for which (17) holds
**Output:** $\nabla F(x) \in \mathbb{R}^d$

**1 for** $j \in [d]$ **do**

**2** $\quad$ Sample random unit vector $z_j \in \mathbb{S}^{d-1}$.

**3** $\quad$ $v_j \leftarrow (F(x + \alpha z_j) - F(x))/\alpha$.

**4** Let $w$ be the solution to the linear system $\{\langle w, z_j \rangle = v_j\}_{j \in [d]}$.

**5 return** $w$

---

and $\alpha|\langle w_i, z \rangle| \leq \alpha \cdot \|w_i\|$, so $\langle w_i, x + \alpha z \rangle - b_i$ and $\langle w_i, x \rangle + b_i$ have the same sign. As a result, if $S \subseteq [k]$ denotes the indices $i$ for which $\langle w_i, x \rangle - b_i > 0$, then

$$\frac{F(x + \alpha z) - F(x)}{\alpha} = \left\langle \sum_{i \in S} s_i w_i, z \right\rangle = \langle \nabla F(x), z \rangle.$$

If $\{z_1, \ldots, z_j\}$ are a collection of Haar-random unit vectors, they are linearly independent almost surely, in which case the linear system in Step 4 of GETGRADIENT has a unique solution, namely $\nabla F(x)$. $\qquad \square$

In order to use GETGRADIENT to construct the vectors $\nabla_L(I)$, we require estimates for $\alpha$ in Lemma 4.9. In the following lemma we show that with high probability over the randomness of $L$, if an interval $I$ completely lies within a linear piece of $F|_L$, then we can bound how small we must take $\alpha$ to query the gradient of $F$ at the midpoint of that interval.

**Lemma 4.10.** *Let $L$ be a Gaussian line. With probability at least $1-\delta$ over the randomness of $L$, the following holds: in the partition $[-\tau, \tau] = I_1 \cup \cdots \cup I_m$ in Theorem 4.8, for any $I_\ell$ which entirely lies within a linear piece of $F|_L$, GETGRADIENT$(t_{\mathrm{mid}}, \alpha)$ correctly outputs $\nabla_L(I_\ell)$, where $x_{\mathrm{mid}}$ is the midpoint of the interval $\mathcal{I}_\ell \subset L$ that corresponds to interval $I_\ell \subset \mathbb{R}$ and $\alpha = \frac{\delta \cdot r}{4k\sqrt{d} + O(k\sqrt{\log(k/\delta)})}$ (where $r$ is defined in (12)).*

*Proof.* Denote $L = \{x_0 + t \cdot v\}_{t \in \mathbb{R}}$. Let $t_{\mathrm{mid}} \in \mathbb{R}$ denote the value corresponding to $x_{\mathrm{mid}} \in \mathbb{R}^d$ on the line $L$. By Lemma 2.7 and a union bound over $[k]$, we have that

$$|\langle w_i, v \rangle| \geq \frac{\delta \|w_i\|}{2k\sqrt{d} + O(k\sqrt{\log(k/\delta)})} \quad \text{for all } i \in [k]$$

with probability at least $1 - \delta$ over the randomness of $v \in \mathbb{S}^{d-1}$. Now take any interval $I_\ell$ which entirely lies within a linear piece of $F|_L$. Because $t_{\mathrm{mid}}$ is the midpoint of $I_\ell$, it is at least $r/2$ away from any critical point of $F|_L$. In particular, $|\langle w_i, x_{\mathrm{mid}} \rangle - b| \geq (r/2) \cdot |\langle w_i, v \rangle| \geq (r/2) \cdot \frac{\delta \|w_i\|}{2k\sqrt{d} + O(k\sqrt{\log(k/\delta)})}$, so we can take $\alpha = \frac{\delta \cdot r}{4k\sqrt{d} + O(k\sqrt{\log(k/\delta)})}$ and invoke Lemma 4.9. $\qquad \square$

Putting these ingredients together, we obtain the following algorithm, GETNEURONS for producing a collection of neurons that can be used to approximate $F$.

We prove correctness of GETNEURONS in the following lemma:

**Lemma 4.11.** *For any $\epsilon, \delta > 0$, GETNEURONS$(\epsilon, \delta)$ makes $\mathrm{poly}(k, d, R, B, 1/\epsilon, \log(1/\delta))$ queries and outputs a list $\mathcal{S}$ of pairs $(w, b)$ for which there exist $\{\pm 1\}$-valued coefficients $\{a_{w,b}\}_{(w,b) \in \mathcal{S}}$ as well as a vector $w^*$ and a scalar $b^*$ such that*

$$\left\| F - \langle w^*, \cdot \rangle - b^* - \sum_{(w,b) \in \mathcal{S}} a_{w,b} \cdot \sigma(\langle w, \cdot \rangle - b) \right\| \leq \epsilon.$$

*Proof.* By Lemma 4.10, the choice of $\alpha$ in GETNEURONS is sufficiently small that for $x_j$ the midpoint of any interval which is entirely contained within a linear piece of $F|_L$, GETGRADIENT$(x_j, \alpha)$ succeeds by Lemma 4.9. So the estimates $\widehat{\nabla}$ and $\widehat{b}$ are exactly correct for all intervals that are entirely

---

**Algorithm 3:** GETNEURONS($\epsilon, \delta$)

---

**Input:** Accuracy $\epsilon > 0$, confidence $\delta > 0$
**Output:** List $\mathcal{S}$ of pairs $(w, b)$ (see Theorem 4.8 for guarantee)

1   $\mathcal{S} \leftarrow \emptyset$.
2   Sample Gaussian line $L$.
3   $\Delta \leftarrow (\epsilon / \mathfrak{P}_{k,R,B,\log(1/\delta)})^{9/2}$.          // Theorem 4.8
4   $\alpha \leftarrow \frac{\delta \cdot r}{4k\sqrt{d} + O(k\sqrt{\log(k/\delta)})}$.          // Lemma 4.10
5   Define $r, \tau$ according to (12), (13).
6   Partition $[-\tau, \tau]$ into disjoint intervals $I_1, \ldots, I_m$ of length $r$.
7   **for** *all* $j \in [m]$ **do**
8      $x_j \leftarrow$ midpoint of the interval $\mathcal{I}_j \subset L$ that corresponds to $I_j \subset \mathbb{R}$.
9      $\widehat{\nabla}_L(I_j) \leftarrow$ GETGRADIENT($x_j, \alpha$).
10     $\widehat{b}_L(I_j) \leftarrow$ GETBIAS($L, I_j$).
11   **for** *all pairs of distinct* $i, j \in [m]$ **do**
12     $(v_j, b_j) \leftarrow (\widehat{\nabla}_L(I_i) - \widehat{\nabla}_L(I_j), \widehat{b}_L(I_i) - \widehat{b}_L(I_j))$.
13     **if** $(v_j, b_j)$ *satisfies the bounds in* (14) **then**
14        Add $(v_j, b_j)$ to $\mathcal{S}$.
15   **return** $\mathcal{S}$.

---

contained within a linear piece of $F|_L$. By the proof of Theorem 4.8, these are the only intervals for which we need $\nabla_L(I)$ and $b_L(I)$ in order for $\mathcal{S}$ to contain enough neurons to approximate $F$ by some linear combination to $L_2$ error $\epsilon$. $\qquad\square$

### C.4   Linear Regression Over ReLU Features

It remains to show how to combine the neurons produced by GETNEURONS to obtain a good approximation to $F$. As Theorem 4.8 already ensures that some linear combination of them suffices, we can simply draw many samples $(x, F(x))$ for $x \sim \mathcal{N}(0, \text{Id})$, form the feature vectors computed by the neurons output by GETNEURONS, and run linear regression on these feature vectors.

Formally, let $\mathcal{S}$ denote the set of pairs $(w, b)$ guaranteed by Theorem 4.8. We will denote the $w$'s by $\{\widehat{w}_j\}$ and the $b$'s by $\{\widehat{b}_j\}$. Consider the following distribution over feature vectors computed by the neurons in $\mathcal{S}$:

**Definition C.1.** *Let $\mathcal{D}'$ denote the distribution over $\mathbb{R}^{|\mathcal{S}|+d+1} \times \mathbb{R}$ of pairs $(z, y)$ given by sampling $x \sim \mathcal{N}(0, \text{Id})$ and forming the vector $z$ whose entries consist of all $\sigma(\langle \widehat{w}_j, x \rangle - \widehat{b}_j)$ as well as the entries of $x$ and the entry 1, and taking $y$ to be $F(x)$ for the ground truth network $F$ defined in (11).*

*We will also need to define a truncated version of $\mathcal{D}'$: let $\mathcal{D}$ denote $\mathcal{D}'$ conditioned on the norm of the $|\mathcal{S}|+1$ to $|\mathcal{S}|+d$-th coordinates having norm at most $M \triangleq \sqrt{d} + O(\sqrt{\log(1/\delta)})$, which happens with probability at least $1 - \delta$ over $\mathcal{D}'$.*

Our algorithm will be to sample sufficiently many pairs $(z, y)$ from $\mathcal{D}'$ (by querying $F$ on random Gaussian inputs) and run ordinary least squares. This is outlined in LEARNFROMQUERIES below.

To show that regression-based algorithm successfully outputs a network that achieves low population loss with respect to $F$, we will use the following standard results on generalization.

**Theorem C.2.** *For $\mathcal{D}$ a distribution over $\mathcal{X} \times \mathcal{Y}$ and $\ell : \mathcal{Y} \times \mathcal{Y} \to \mathbb{R}$ a loss function that is $L$-Lipschitz in its first argument and uniformly bounded above by $c$. Let $\mathcal{F}$ be a class of functions $\mathcal{X} \to \mathcal{Y}$ such that for any $f \in \mathcal{F}$ and pairs $(x_1, y_1), \ldots, (x_n, y_n)$ drawn independently from $\mathcal{D}$, with probability at least $1 - \delta$,*

$$\mathbb{E}_{(x,y) \sim \mathcal{D}}[\ell(f(x), y)] \leq \frac{1}{n}\sum_i \ell(f(x_i), y_i) + 4L \cdot \mathcal{R}_n(\mathcal{F}) + 2c \cdot \sqrt{\frac{\log(1/\delta)}{2n}},$$

*where $\mathcal{R}_n(\mathcal{F})$ denotes the Rademacher complexity of $\mathcal{F}$.*

---

**Algorithm 4:** LEARNFROMQUERIES($\epsilon, \delta$)

**Input:** Accuracy $\epsilon > 0$, confidence $\delta > 0$
**Output:** One hidden-layer network $\widetilde{F} : \mathbb{R}^d \to \mathbb{R}$ for which $\|F - \widetilde{F}\| \le O(\epsilon)$

**1** $\mathcal{S} = \{(\widehat{w}_j, \widehat{b}_j)\} \leftarrow$ GETNEURONS($\epsilon, \delta$).
**2** Draw samples $(z_1, y_1), \ldots, (z_n, y_n)$ from $\mathcal{D}$         // Definition C.1
**3** Let $\widetilde{v}$ be the solution to the least-squares problem (19). Let $\widetilde{b}$ denote the last entry of $\widetilde{v}$, and
    let $\widetilde{w}$ denote the vector given by the $d$ entries of $\widetilde{v}$ prior to the last.
**4** Form the network $\widetilde{F}(x) \triangleq \sum_j \widetilde{v}_j \sigma(\langle \widehat{w}_j, x\rangle - \widehat{b}_j) + \langle \widetilde{w}, \cdot\rangle - \widetilde{b}$.
**5** **return** $F$.

---

**Theorem C.3.** *If $\mathcal{X}$ is the set of $x$ satisfying $\|x\| \le X$, and $\mathcal{F}$ is the set of linear functions $\langle w, \cdot\rangle$ for $\|w\| \le W$, then $\mathcal{R}_n(\mathcal{F}) \le XW/\sqrt{n}$.*

As these apply to bounded loss functions and covariates, we must first pass from $\mathcal{D}'$ to $\mathcal{D}$ and quantify the error in going from one to the other:

**Lemma C.4.** *For $f$ satisfying $\mathbb{E}_{(z,y)\sim\mathcal{D}'}[(f(z) - y)^2] \le \epsilon^2$, we have*

$$\left| \mathop{\mathbb{E}}_{(z,y)\sim\mathcal{D}'}[(f(z) - y)^2] - \mathop{\mathbb{E}}_{(x,y)\sim\mathcal{D}}[(f(z) - y)^2] \right| \le O(\epsilon^2). \tag{18}$$

*Proof.* Let $Z$ denote the probability that a random draw from $\mathcal{D}'$ lies in the support of $\mathcal{D}$ so that $Z \ge 1 - \delta$; denote this event by $\mathcal{E}$. Then we can write $\mathbb{E}_{(z,y)\sim\mathcal{D}}[(f(z) - y)^2]$ as $\frac{1}{Z}\mathbb{E}_{(z,y)\sim\mathcal{D}'}[(f(z) - y)^2 \cdot \mathbb{1}[z \in \mathcal{E}]]$ and rewrite the left-hand side of (18) as

$$\left| \left(1 - \frac{1}{Z}\right) \cdot \mathop{\mathbb{E}}_{(z,y)\sim\mathcal{D}'}[(f(z) - y)^2 \cdot \mathbb{1}[z \in \mathcal{E}]] + \mathop{\mathbb{E}}_{(z,y)\sim\mathcal{D}'}[(f(z) - y)^2 \cdot \mathbb{1}[z \notin \mathcal{E}]] \right|.$$

Note that $|1 - 1/Z| \le 2\delta \le 1$ for $\delta$ sufficiently small, from which the claim follows. $\qquad\square$

We are now ready to prove the main theorem of this section:

**Theorem 4.12.** *Let $\mathcal{S}$ denote the list of pairs $(\widehat{w}_j, \widehat{b}_j)$ output by GETNEURONS($\epsilon, \delta$). Sample $(z_1, y_1), \ldots, (z_n, y_n)$ from $\mathcal{D}$ for $n = \mathrm{poly}(k, R, B, 1/\epsilon, d, \log(1/\delta))$. With probability at least $1 - O(\delta)$ over the randomness of GETNEURONS and the samples, the following holds. Define*

$$\widetilde{v} \triangleq \arg\min_{\|v\| \le W} \sum_{i=1}^{n} (\langle v, z_i\rangle - y_i)^2, \text{ for } W \triangleq \sqrt{\tau/r} + k(R + B), \tag{19}$$

*let $\widetilde{b}$ denote the last entry of $\widetilde{v}$, and let $\widetilde{w}$ denote the vector given by the $d$ entries of $\widetilde{v}$ prior to the last. Then the one hidden-layer network $\widetilde{F}(x) \triangleq \sum_j \widetilde{v}_j \sigma(\langle \widehat{w}_j, x\rangle - \widehat{b}_j) + \langle \widetilde{w}, \cdot\rangle - \widetilde{b}$ satisfies $\|F - \widetilde{F}\| \le O(\epsilon)$.*

*Proof.* Note that over the support of $\mathcal{D}$ we have that the square loss $\ell : \mathcal{Y} \times \mathcal{Y} \to \mathbb{R}$ is uniformly bounded above by $(MkR + kB)^2$ and is $L = O(M \cdot k \cdot R + k \cdot B)$-Lipschitz. Finally, note that for $z$ in the support of $\mathcal{D}$,

$$\|z\|^2 = 1 + M^2 + 2M^2 \sum_j (\|\widehat{w}_j\|^2 + \widehat{b}_j^2)$$

$$\lesssim (M^2 \tau/r) \cdot (k^2 R^2 + \Delta^2 k^4 R^2 \log(k/\delta) + k^2 B^2) \triangleq X^2. \tag{20}$$

where $\tau, r$ are defined in Theorem 4.8 and we used (14) and Step 14 in GETNEURONS to bound $\|\widehat{w}_j\|$ and $|\widehat{b}_j|$.

By the guarantee on GETNEURONS given by Lemma 4.11, we know that there is a vector $v^* \in \{\pm 1\}^{|\mathcal{S}|} \times B^d(kR) \times [-kB, kB]$ which achieves $\epsilon^2$ squared loss with respect to $\mathcal{D}'$. Note that

$$\|v^*\| \le |\mathcal{S}|^{1/2} + k(R + B) = \sqrt{\tau/r} + k(R + B) \triangleq W. \tag{21}$$

By Lemma C.4, $v^*$ achieves $O(\epsilon^2)$ squared loss with respect to $\mathcal{D}$. As the random variable $(\langle v^*, z \rangle - y)^2$ for $(z, y) \sim \mathcal{D}$ is bounded above by

$$(\|v^*\|\|z\| + |y|)^2 \lesssim \text{poly}(k, R, B, 1/\epsilon, M),$$

for $n \geq \text{poly}(k, R, B, 1/\epsilon, M)$ we have that the empirical loss of $v^*$ on $(z_1, y_1), \ldots (z_n, y_n)$ is $O(\epsilon^2)$, and therefore that of the predictor $\widetilde{v}$ is $O(\epsilon^2)$.

By applying Theorem C.3 with (20) and (21), we find that the Rademacher complexity $\mathcal{R}_n(\mathcal{F})$ of the family of linear predictors over $\|z\| \leq X$ and with norm bounded by $W$ is $C/\sqrt{n}$ for $C$ which is polynomial in $k$, $R$, $B$, $1/\epsilon$, $d$, $\log(1/\delta)$, from which the theorem follows by Theorem C.2. $\qquad\square$