# OpenReview forum: "Efficiently Learning One Hidden Layer ReLU Networks From Queries"
_NeurIPS.cc/2021/Conference — NeurIPS 2021 Poster_

### Official Review · Reviewer_zwJ7 · 2021-07-10

**Rating:** 7
**Confidence:** 4

**Summary:**

The paper considers the problem of learning depth-2 ReLU networks w.r.t. Gaussian inputs, where the learner has query (black-box) access to the target network. The problem of learning depth-2 networks (without queries) has been extensively studied in recent years. All known efficient algorithms require assumptions on both the input distribution and the network’s weights (e.g., that the weights matrix has full rank). Moreover, the known efficient algorithms for depth-2 ReLU networks do not handle bias terms. In this paper the authors give a first efficient algorithm for learning depth-2 ReLU networks with bias w.r.t. the Gaussian distribution using black-box access. There are no assumptions on the weights, except for having a bounded norm. The motivation for the problem is two-fold: (1) the model of PAC learning with membership queries is well-studied and natural; (2) there is an interest in model-extraction attacks in the security and privacy communities. Prior works on model-extraction either gave algorithms without theoretical guarantees, or efficient algorithms under strong assumptions.

The proposed algorithms follows the following ideas. For a random line L consider the restriction of the target network F. This restriction is a piecewise linear univariate function, such that each critical point corresponds to a neuron in F. This property can be exploited in order to recover the neurons. However, there are some major issues to overcome in order to implement this idea efficiently. For example, since we do not have any assumptions on the weights, then the critical points might be very close to each other. The authors solve this issue by showing that this situation occurs only if there is a cluster of similar neurons, in which case we can replace them with a small sub-network and do not have to recover all the neurons in the cluster.


**Limitations And Societal Impact:**

Yes

**Main Review:**

The model of learning with queries has gained interest both from the theoretical and from the practical viewpoints. The authors show that for the case of depth-2 ReLU networks, black-box access allows us to achieve learners with improved guarantees. I think that the problem considered in the paper is natural and that the contribution is significant. I have some comments on the presentation, which I discuss below.

The authors mention that obtaining polynomial-time algorithms for learning depth-2 networks without assumptions on the weights (and without queries) is an open problem. I think that mentioning the known hardness results for the problem would help understanding the context. Statistical queries lower bounds for learning depth-2 networks on the Gaussian distribution is given in [Ilias Diakonikolas, Daniel M Kane, Vasilis Kontonis, and Nikos Zarifis. Algorithms and sq lower bounds for pac learning one-hidden-layer relu networks] and in [Surbhi Goel, Aravind Gollakota, Zhihan Jin, Sushrut Karmalkar, and Adam Klivans. Superpolynomial lower bounds for learning one-layer neural networks using gradient descent]. Also, hardness of improperly learning depth-2 networks (with any algorithm) over another product distribution is given in [Amit Daniely and Gal Vardi. From Local Pseudorandom Generators to Hardness of Learning]. Hence, by these hardness results, obtaining a result similar to Thm 1.1 but without queries is probably impossible. It implies that in the case of depth-2 networks learning with queries seems to be strictly stronger than learning from random samples.

Example 1.2: I suspect that there is an error here. For example, if x_2 = 4/Lambda and 0<x_1<1 then F(x_1,x_2) = x_1+4 + 3x_1+4  - 2(-x_1+4) = 6x_1 \neq 0.

In line 117-119 it says: “we also emphasize that our results are the first theoretical guarantees for learning one hidden-layer networks with bias terms in any learning model, including PAC learning from samples”. Note that JSA15 also allows a bias term (although the assumptions on the activation function there rules out the ReLU function).

Lemma 4.11 argues that F can be approximated by a combination of the neurons returned by GetNeurons. However, by Thm 4.8 it is not sufficient to use these neurons, since there is also a term <w*,\cdot> and a term b*. Am I missing something here?

Section 4.4: First, the distributions D’ and D are not defined (they are only defined in the appendix). Also, by observing the appendix, I see that the distributions D and D’ are such that z has dimension |S|+d+1 and hence it seems that the linear regression of algorithm 4 finds also the required w* and b* that I mentioned in the previous comment. However, even in the appendix, in Thm 4.12 the returned function \tilde{F} uses only the neurons in S and does not use w* and b*.

More comments:
- Lemma 2.6: There exists C s.t. for every \nu …
- Lemma 2.7: t and c are introduced but not used.
- Line 203 (and line 266 in the supplementary): “b* + =“.
- Lemma 4.4: c is defined by not used.
- Algorithm 4: “// Definition ??” and “problem (??)”.

Overall, I think that the contribution is significant enough to merit acceptance.

**Post rebuttal update**
Thank you for the response. After reading the other reviews and the responses, I maintain my recommendation for acceptance.

**Time Spent Reviewing:**

6

---

> ### Author Response · Authors · 2021-08-10
> **Thanks for the detailed feedback!**
>
> Thanks very much for taking the time to read our submission closely and for the very helpful feedback! We will make sure to address the typos mentioned in "More comments" and focus here on the comments in the body of the review.
>
> First, thanks for the suggestion about including the known hardness results for PAC learning depth-2 networks! In the final version we will make sure to include a discussion about these and mention the implication that you note, namely that our result gives separation between query and PAC learning for depth-2 networks.
>
> Regarding example 1.2, sorry for the typo! The coefficient of 3 in the second neuron should be -3 so that the coefficients in front of the $x_1$'s cancel. In other words, the function should be $F(x,y) = \\sigma(x_1+\\Lambda\\cdot x_2) + \\sigma(-3x_1+\\Lambda\\cdot x_2) - 2\\sigma(-x_1+\\Lambda\\cdot x_2)$. So for instance, if $x_2 = 4/\\Lambda$, then $F(x_1,x_2) = \\sigma(x_1+4) + \\sigma(-3x_1+4) - 2\\sigma(-x_1+4)$. If $-1 < x_1 < 1$, then all neurons are activated, so this equals $(x_1+4) + (-3x_1+4) - 2(-x_1+4) = 0$.
>
> Regarding lines 117-119, we apologize for overlooking JSA15 and will modify that sentence accordingly to acknowledge JSA15 and point out that, as you say, their result does not pertain to ReLU activations.
>
> Finally, regarding your comments about Lemma 4.11/GetNeurons and Thm 4.12/LearnFromQueries, you are correct that there are a few typos. Lemma 4.11 should instead say "there exist $\\{\pm 1\\}$-valued coefficients $\\{a_{w,b}\\}$ as well as a vector $w^*$ and scalar $b^*$ such that $\\|F - \\langle w^*,\\cdot\\rangle - b^* - \\sum a_{w,b}\\cdot\\sigma(\\langle w,\\cdot\\rangle - b)\\| \\le \\epsilon$." Thm 4.12 and the pseudocode for LearnFromQueries should similarly be modified to include a $\\langle w^*,\\cdot \\rangle$ and $b^*$ term. In any case, the statements of these results will hold after these changes because we are simply running linear regression in the $(|S| + d + 1)$-dimensional feature space consisting of $\\mathbb{R}^{d+1}$ (for the linear and bias terms) together with the $|S|$ features corresponding to the neurons obtained by GetNeurons.

---

### Official Review · Reviewer_ULd9 · 2021-07-14

**Rating:** 8
**Confidence:** 3

**Summary:**

This paper provides a polynomial-time algorithm for learning neural networks (NN) with one hidden layer through queries. The result holds for NN with bounded weights and biases under Gaussian distributions.

The authors define a notion of closeness between neurons that allows strong assumptions on the weights (which were used in prior work and didn't translate to the overparametrized setting) to be dropped. Indeed, if a set of neurons are "close" enough, they can be approximated by two neurons or a linear function. This allows the learning algorithm to efficiently represent the target NN, which could be overparametrized.

At a high level, the algorithm recovers neurons by sampling a Gaussian line, separating it into suitably small intervals and computing the gradient (through queries) at their midpoints to recover the neurons' weight vectors.

**Limitations And Societal Impact:**

Yes

**Main Review:**

I believe this is a strong paper. It is interesting in its own right, and, as the authors mention, the question is particularly relevant given the practical threat of model extraction in adversarial machine learning. Moreover, the theoretical results on the structural properites of NN (neuron closeness, learning a NN with biases) and the fact that the algorithm works in an overparametrized regime are of independent interest.

The paper is well-written and has a great structure/organization (especially for the outline of the technical results, which gives the reader a clear idea of the proof without being overwhelmed by technical details). The related work section is detailed and it is clear how the work differs from results in the literature (and how previous approaches would fail in the given problem setting). However, a major flaw is the lack of a discussion/conclusion section, which would need to be added were the paper to be accepted.

Finally, the paper could greatly benefit from an experimental section that implements the algorithm and evaluates it on synthetic and real-world data.

Minor comments:
- l.27-30: Please add the distributional assumption in the theorem statement
- l.46-48: Please define x_0 and v (\in \R^d)

Suggestions:
- l.19-22: It would be great to have a table of different NN learning algorithms (distributional and structural assumptions, runtime, etc.) to compare recent work with this paper
- l.124-125: N(0,I_d) perhaps add that this is WLOG, as we can centre + normalize the training data
- l.133: it would be a good idea to add some text here introducing the lemmas/facts, where they will be used, and stating that the proofs are in the appendix
- l.134,138: maybe add "we denote by ... the operator norm, and by ... the Frobenius norm" at the beginning of the preliminary section (they haven't been introduced). Also, define S and Haar-randomness
- Lemma 3.8: perhaps define B and R before "If signs s_1,..."
- l.297-298 and Algorithm 3: could you please add a few sentences explaining what happens when an interval doesn't lie within a linear piece of F_L, and why if (v_j, b_j) can be dropped if they don't satisfy equation 4 (line 13 of Algorithm 3)?

Typos:
- l.143: "the supplement" -> "the supplementary material"
- l. 169: "satisfies triangle inequality" -> "satisfies the triangle inequality"
- l.203 typo "+" in the definition of b^*?
- l.289: "several sufficiently small" -> "for several sufficiently small"
- l.306: "we prove correctness" -> "we prove the correctness"
- Algorithm 4: Definition and LS problem references are "??"

**Time Spent Reviewing:**

5

---

> ### Author Response · Authors · 2021-08-10
> **Thanks for the helpful suggestions!**
>
> Thanks very much for taking the time to read our submission closely and for the positive feedback and helpful suggestions! To address the main points touched upon in this review:
> - We will make sure to add a concluding section summarizing and contextualizing our contributions as well as suggesting future directions.
> - We also strongly agree that our algorithm should be empirically evaluated and intend to pursue this in future work.
>
> Lastly, we greatly appreciate the reviewer’s helpful minor comments/suggestions/typo fixes and will be sure to incorporate them into the final version.

---

### Official Review · Reviewer_FZeD · 2021-07-16

**Rating:** 7
**Confidence:** 3

**Summary:**

Present the first provably polynomial-time algorithm for learning unknown one-layer neural networks when given black-box access to the unknown network.

**Ethical Concerns:**

Yes, see $\textbf{Broader Impact}$

**Ethics Review Area:**

["Inappropriate Potential Applications & Impact  (e.g., human rights concerns)", "Responsible Research Practice (e.g., IRB, documentation, research ethics)"]

**Limitations And Societal Impact:**

$\textbf{Limitations}$

The authors have not discussed the limitations in an explicit section, although there is some discussion within the paper. I think the limitations could have been discussed more explicitly.

$\textbf{Broader Impact}$

There is no discussion about the potential negative societal impacts of this work. Given the nature of this work, I think there is certainly potential for negative impact (e.g. model extraction attacks, as stated by the authors but not discussed). I think a section dedicated to this discussion is necessary given the content of the paper, particularly talking about how to mitigate any potential negative impact.



**Main Review:**

$\textbf{Originality}$

To my knowledge, the algorithm presented is novel, and it is clear how this work both differs from and builds upon previous contributions to this topic.

$\textbf{Quality}$

The theorems are clear and supported with sound proofs.

$\textbf{Clarity}$

The submission is clearly written and easy to follow.

$\textbf{Significance}$

Efficiently learning an unknown one-layer network has been an open question with a lot of active research. I believe that such an algorithm is certainly significant.

**Needs Ethics Review:**

Yes

**Time Spent Reviewing:**

8

---

> ### Author Response · Authors · 2021-08-10
> **Thanks for the helpful feedback on limitations/broader impact!**
>
> Thanks very much for taking the time to understand our submission and for the positive review!
>
> In the final version, we will make sure to add a section explicitly spelling out the limitations, the primary one being that our result is limited to depth-two networks.
>
> As for broader impact, we agree that taken to its logical conclusion, this line of work on learning neural networks from black-box access does pose a risk, for instance, that proprietary models offered via publicly-hosted APIs may be stolen by attackers who are only able to interact with the models through queries. The attackers might then use the extracted parameters of the model to learn sensitive information about the data the model was trained on, or perhaps to construct adversarial examples. All that being said, we believe that one motivation for developing theorems about query learning is that the theory that gets built along the way can help suggest ways of *provably* mitigating such threats. For instance, it’s conceivable that one can prove information-theoretic or computational hardness for such problems if appropriate levels and kinds of noise are injected into the responses to the queries. Furthermore, query complexity *lower bounds* can inform how many accesses an API should allow any given user. We will make sure to spell out all of this more explicitly in a dedicated “Broader Impact” section.

---

### Official Review · Reviewer_e3xE · 2021-07-19

**Rating:** 6
**Confidence:** 4

**Summary:**

In this work, the authors study the problem of learning one hidden layer neural network with ReLU activations with $k$ neurons. Their objective is to output a ReLU network $F'$, such that $\|F-F'\|_2^2\leq \epsilon$, where $F$ is the unknown network. They assume that the underlying distribution is Gaussian, and that they have query access to the network. A query is a function of the form $q(x)=F(x)$, that is given an $x$, the query returns the value of $F(x)$. In this work, the only assumption is that $F(x)$ is bounded with respect to the $\|\cdot\|_2$. Other works have required either the vectors $\vec w$ to be linear independent or the weights to be non-negative. The assumptions are required in general, as linear independence is required for parameter estimation, and there exists Statistical Queries hardness on getting $\|F-F'\|_2^2\leq \epsilon$ using polynomial many queries without the assumption of non-negative weights. By assuming a more powerful model with queries they overcome the hardness.

Their algorithm restricts $F$ to a random line $L$ and then uses queries to estimate approximations to the gradients and get the critical points (points where a neuron is activated, i.e., changes sign) of $F$ restricted to the line $L$, and then they can estimate the $w_i$, and the $b_i$. One problem of this approach is that fails if some neurons are close to each other. To overcome this, the authors prove that if some neurons are close to each other, then they can be approximate by a linear combination of at most two neurons. The algorithm uses polynomial samples and queries with respect the parameters $d,max\|w\|,max(b),1/\epsilon,k$.

**Limitations And Societal Impact:**

This work does not seem to have any negative societal impact.

**Main Review:**

The authors study an interesting problem for the community of statistical learning and provide a polynomial learning algorithm with mild assumptions. The authors overcome several difficulties of this problem by using queries to the unknown function. I find the idea of searching in a Gaussian line for critical points very interesting which might also be of independent interest.

The main weakness of this work is that the queries that are needed are comparable with the overall sample complexity. I would guess that the queries that are needed for this problem, should be less (i.e., logarithmic) and/or with fewer dependencies on the parameters than the ones required by the samples.

Other:
1. $W$ changes to $R$ in the following sections.
2. Instead of having bounds on the norm of the vectors and the parameters, wouldn't a bound on the $\|F\|_2$ norm suffice?
3. Lemma 3.3 is difficult to parse.
4. Line 203 $b^*+$ should be $b^*$
5. I would suggest changing the structure of Theorem 4.8 to make it easier to parse.
6. Algorithm 4 has broken references.
7, Authors should clarify which of the previous work did parameter estimation vs function estimation.

**Time Spent Reviewing:**

5

---

> ### Author Response · Authors · 2021-08-10
> **Thanks for the great questions!**
>
> Thanks for taking the time to review our submission closely and for raising very interesting questions! Here we address the main comments raised, though we will also make sure to address the typos and writing suggestions in the final version of the paper.
>
> The primary complaint was that our query complexity bound is no better than the sample complexity for PAC learning, e.g. we still only get $\text{poly}(1/\epsilon)$ rather than $\log(1/\epsilon)$. To summarize our discussion below:
> 1. $\log(1/\epsilon)$ is impossible (see below for a counterexample)
> 2. For PAC learning from samples, $\text{poly}(1/\epsilon)$ sample complexity is not achievable by any known algorithm for *general* two-layer neural networks that runs polynomial in the ambient dimension. In contrast, for query learning, we get a query complexity and runtime bound that is polynomial in all parameters.
>
> As this was cited in the review as the main weakness, we hope in light of this that you decide to increase your score.
>
> To see 1), consider the two-layer network over one-dimensional input given by $F(x) = x + \sigma(x - c) + \sigma(x - c - \epsilon^{2/3}) - \sigma(2x - 2c - \epsilon^{2/3})$ for some unknown $c\in[-1,1]$. This basically looks like a straight line but with a small “bump” over the interval $[c,c+\epsilon^{2/3}]$. In particular, the L2 distance between this function and the function $F(x) = x$ is $\Theta(\epsilon)$, so we really need to learn where the bump is. But if the learner queries any point $x$ outside of $[c,c+\epsilon^{2/3}]$, she only sees $x$ as the output. This implies that at least $\Omega(1/\epsilon^{2/3})$ queries are necessary. One can still ask whether one can get slightly better polynomial dependence on $\epsilon$ via query learning than via PAC learning from samples, but this is a rather technical question that is a bit out of the scope of what we sought to achieve in our submission. As for the remaining parameters like $d$ and $k$, one cannot hope to do better in the query setting than in the sample setting because you still need at least $dk+k$ queries just by a parameter count. We did not attempt to optimize our polynomial dependence on these parameters but agree that it is a worthwhile future direction to pursue.
>
> As for 2), the only known algorithm for *general* two-layer neural networks that runs polynomial in ambient dimension is [CKM20], and they incur an exponential dependence on $1/\epsilon$.
>
> The reviewer also asked why not just assume L2 norm of $F$ is bounded rather than that the weight vectors/biases are bounded. Here's a proof that one cannot obtain finite query complexity bounds under such an assumption. Consider the two-layer network with one-dimensional input given by $F(x) = C\cdot (\sigma(x - c) + \sigma(x - c - \delta) - \sigma(2x - 2c - \delta))$, which just looks like a bump over the interval $[c,c+\delta]$ and is zero elsewhere. This function has L2 norm $O(C\cdot \delta^{3/2})$, so take $C = \delta^{-3/2}$ so it has unit $L_2$ norm. Because we are only assuming the $L_2$ norm is bounded and not that the weights are bounded, we are free to pick $\delta$ arbitrarily, and in order to learn this $F$, we really have to identify where the bump is. But even if we knew $c$ were bounded in an interval, say over $[-1,1]$, we’d need at least $\Omega(1/\delta)$ queries to find the bump, and we can take $\delta$ to be arbitrarily small so that the query complexity is unbounded.
>
> Nevertheless, both of the above questions were very interesting, and we will be sure to add a discussion of these points to the intro!
>
> Finally, regarding the question about which previous works did parameter versus function estimation, we will add a note in the introduction clarifying this. Of the three most relevant works, namely [JCB+20], [CJM20], [MSDH19], the works of [JCB+20] and [MSDH19] study function estimation (in the security parlance, “fidelity extraction”), while [CJM20] gives an algorithm that empirically achieves parameter estimation for underparametrized architectures (see Table 1 of [CJM20]).

---

### Review · Ethics_Reviewer_33f4 · 2021-07-30

**Recommendation:**

The paper should include discussion of the potential negative implications of this research and the ways in which they believe these outcomes can be avoided. Open research into model extraction attacks is clearly necessary in order to improve security and mitigate risk. However, these questions are not - at present - engaged with.

My review is in agreement with the review submitted by reviewer FZeD.

**Ethical Issues:**

Yes

**Ethics Review:**

This paper looks at techniques that are part of research into model extraction attacks. As the researchers note in the introduction, this area is important for the privacy and security community because effective attacks have the potential to undermine the security of ML systems. In the light of this the research clearly raises safety and security concerns (see ethical guideline Q.2). As such, the paper needs to contain a section discussing these potential effects and potential forms of mitigation.

---

### Review · Ethics_Reviewer_iCqz · 2021-08-12

**Recommendation:**

See above. Authors should include explicit explanation of why this work is beneficial in better understanding and ultimately preventing model extraction attacks. Even if this paper only lays the foundation for such work.

**Ethical Issues:**

Yes

**Ethics Review:**

As Reviewer FZeD notes, this work could be understood as contributing to model extraction attack capabilities. This is not framed as threat modeling or a mitigation strategy. (My technical knowledge is limited. But, it seems like this could be read as a bit of a "how-to" manual.)

---

### Author Response · Authors · 2021-08-29
**Response to ethics reviews**

We sincerely apologize for the delayed response, unfortunately we only noticed the two ethics reviews today! We really appreciate the identification of the issues that we need to address as well as the recommendations for how to move forward. We are encouraged that after our response to reviewer FZeD, ethics reviewer iCqz noted that the additions we mentioned in our response to FZeD would adequately reframe the paper to address the ethical concerns. As stated in that response, we are in complete agreement that the dangers of designing algorithms for model extraction are very real. One of the high-level motivations for our submission is that an important step towards mitigating these dangers is to arm the community with rigorous principles inspired by learning theory for reasoning about when providing query access to a trained model is and is not safe. We will make sure in the final version of the draft to incorporate elements of this discussion so that this point becomes clear and also formulate new directions towards mitigating such attacks, e.g. studying noisy settings where one can prove query complexity lower bounds.

---

### Decision · Program_Chairs · 2021-09-27

**Decision:**

Accept (Poster)

**Comment:**

A solid progress in a well studied line of research